EMBO
Molecular Medicine

# The IL-6 signaling complex is a critical driver, negative prognostic factor, and therapeutic target in diffuse large B-cell lymphoma

Hind Hashwah[1,†], Katrin Bertram[1,†], Kristin Stirm[1], Anna Stelling[1], Cheuk-Ting Wu[1], Sabrina Kasser[1], Markus G Manz[2,3], Alexandre P Theocharides[2,3], Alexandar Tzankov[4] & Anne Müller[1,3,*] [iD]

## Abstract

Interleukin-6 (IL-6) is a growth factor for normal B cells and plasma cell-derived malignancies. Here, we show that the IL-6 signaling pathway is also active in a subset of diffuse large B-cell lymphoma (DLBCL) patients with particularly poor prognosis. Primary DLBCL cells and DLBCL cell lines expressing IL-6R engraft and form orthotopic lymphomas in humanized mice that ectopically produce human IL-6, and in mice reconstituted with a human immune system. We show that a subset of DLBCL cases have evolved mechanisms that ensure constitutive activation of the IL-6 signaling pathway, i.e., the expression of both chains of the IL-6R, the expression of the cytokine itself, and the mutational inactivation of a negative regulator of IL-6 signaling, SOCS1. IL-6 signaling promotes MYC-driven lymphomagenesis in a genetically engineered model, and treatment with the IL-6R-specific antibody tocilizumab reduces growth of primary DLBCL cells and of DLBCL cell lines in various therapeutic settings. The combined results uncover the IL-6 signaling pathway as a driver and negative prognosticator in aggressive DLBCL that can be targeted with a safe and well-tolerated biologic.

**Keywords** cancer immunotherapy; DLBCL mouse models; lymphoma microenvironment; oncogenic STAT3 signaling; personalized treatment
**Subject Categories** Cancer; Immunology; Vascular Biology & Angiogenesis

## Introduction

Diffuse large B-cell lymphoma (DLBCL) is an aggressive malignancy of the mature B cell. DLBCL originates from antigen-exposed B cells that have undergone the germinal center (GC) reaction (Basso & Dalla-Favera, 2015); one of the disease hallmarks is its inter-tumoral heterogeneity. Whereas gene expression profiling has traditionally been used to identify two major subsets of DLBCL that differ in their cell of origin, i.e., the activated B-cell and germinal center B-cell subtypes (ABC- and GCB-DLBCL; Alizadeh *et al*, 2000; Rosenwald *et al*, 2002), more recent studies have distinguished up to four (Schmitz *et al*, 2018) or five (Chapuy *et al*, 2018) different molecular subtypes based on transcriptional and mutational signatures, somatic copy number alterations, and structural variants. These subtypes differ strongly in their response to untailored treatments and survival probability. In particular, the refined stratification based on genetic abnormalities has led to the differentiation of two subsets of ABC-DLBCL, of which one is characterized by (co-occurring) *MYD88* and *CD79B* mutations, extranodal manifestations, a genetic signature of aberrant somatic hypermutation driven by activation-induced cytidine deaminase activity, and a dismal prognosis, whereas the other is characterized by *NOTCH2* and *BCL6* mutations and structural aberrations, respectively, and associated downstream transcriptional signatures, a presumably extrafollicular origin more reminiscent of marginal zone lymphoma, and a comparatively superior prognosis (Chapuy *et al*, 2018; Schmitz *et al*, 2018). Similarly, GCB-DLBCL can be stratified into two subtypes, of which one typically harbors translocations juxtaposing *BCL2* to the *IGH* enhancer in combination with frequent mutations of the chromatin modifiers *KMT2D*, *CREBBP*, and *EZH2*, and inactivating *PTEN* mutations, bears similarities to the genetic landscape of follicular lymphoma and features a poor prognosis, whereas the other is a relatively low-risk subtype with mutations in PI3K-, JAK/STAT-, and MAPK-pathway components and histones (Chapuy *et al*, 2018; Schmitz *et al*, 2018).

The current standard of care for DLBCL, irrespective of subtype, is a combination of chemotherapy and the CD20-targeting antibody rituximab (R-CHOP; Coiffier *et al*, 2002; Basso & Dalla-Favera, 2015). Chimeric antigen receptor T-cell therapy was approved in

1    Institute of Molecular Cancer Research, University of Zurich, Zürich, Switzerland
2    Department of Medical Oncology and Hematology, University Hospital Zurich and University of Zurich, Zürich, Switzerland
3    Comprehensive Cancer Center Zurich, Zürich, Switzerland
4    Institute of Pathology, University Hospital Basel, Basel, Switzerland
     *Corresponding author. Tel: +41 4463 53474; E-mail: mueller@imcr.uzh.ch
     †These authors contributed equally to this work

2017 for relapsed or refractory DLBCL patients, and produces durable remissions in 30–40% of these very difficult-to-treat cases, but is associated with severe side effects (Chow et al, 2018). Other approved or experimental targeted treatments may be suitable for specific subsets of patients. For example, the Bruton tyrosine kinase inhibitor ibrutinib is active in 37% of cases of ABC-DLBCL (Wilson et al, 2015). The susceptibility to ibrutinib is limited to patients with tumors exhibiting both CD79B and MYD88 (L265P) mutations (Wilson et al, 2015), which are characterized by the formation of a multiprotein supercomplex comprising MYD88, TLR9, and the BCR that co-localizes with mTOR on endolysosomes and drives pro-survival NF-κB and mTOR signaling (Phelan et al, 2018). Other experimental compounds that are currently tested in DLBCL patients include lenalidomide, which appears to be preferentially active in the ABC-DLBCL subset of patients (Hernandez-Ilizaliturri et al, 2011; Vitolo et al, 2014), and the proteasome inhibitor bortezomib (Ruan et al, 2011; Offner et al, 2015; Leonard et al, 2017).

Whereas a detailed picture is now emerging of the heterogenous genetic landscape of DLBCL due to huge multi-omics profiling efforts performed on hundreds of patients, the development of suitable mouse models that would allow for predictive pre- or co-clinical testing is woefully lagging behind. Preclinical testing is done using ectopic xenotransplantation models and a small selection of cell lines that are of limited predictive value; primary patient material is notoriously difficult to grow in vitro and will not engraft readily in immunocompromised mouse strains. The available genetic lymphoma models, mostly taking advantage of aberrant MYC or BCL2 overexpression in the B-cell compartment, fail to capture the heterogeneity of the human disease. Here, we show that a genetically humanized mouse strain, the MISTRG mouse, and its derivatives either expressing human IL-6 or reconstituted with a normal human immune system lend themselves to the generation of convenient, rapid-onset orthotopic models that feature tumor engraftment and growth in both lymphoid and non-lymphoid tissues. When combined with in vivo optical imaging system (IVIS) technology, the models allow for the monitoring over time of the tumor burden, tumor dynamics and tissue tropism, clinical symptoms, and treatment responses, not only of cell lines but also of primary patient material. The orthotopic MISTRG model has allowed us to uncover a previously unappreciated dependence of a subset of DLBCL on the IL-6 signaling pathway, which can be exploited therapeutically with a specific monoclonal antibody that is approved for other unrelated indications. Biomarkers that may guide treatment decisions include the tumor cell-intrinsic expression of a functional IL-6 receptor and the constitutive phosphorylation of the downstream transcription factor STAT3, which can be assessed by routine flow cytometric or immunohistochemical testing. In conclusion, we describe here a new pathogenetic pathway that is active and druggable in a subset of high-risk DLBCL patients.

# Results

### DLBCL cell lines engraft in lymphoid and non-lymphoid tissues of MISTRG mice

We have reported recently that the DLBCL cell lines U-2932 (Hashwah et al, 2017) and RC-K8 (Stelling et al, 2018) engraft in

the bone marrow and/or spleen of MISTRG mice within 4 weeks of their intravenous administration. To further refine—and increase the versatility of—this model, we engineered the DLBCL cell lines U-2932, RC-K8, and RIVA to stably express the green fluorescent protein ZsGreen as well as firefly luciferase, which enabled us to visualize their engraftment over time as well as their tissue tropism using IVIS. All three cell lines showed distinct engraftment dynamics, with RC-K8 cells growing more slowly than the other two, and showing an almost exclusive preference for the jaw bone that was not observed with the other cell lines (Fig 1A–C). U-2932 and RIVA cells colonized the hip and long bones of the legs as judged by their IVIS signal (Fig 1A–C). A careful whole-body analysis through a combination of IVIS of explanted organs (representative IVIS images shown in Fig EV1A) with macroscopic inspection at the study endpoint further revealed that all three cell lines rather consistently colonize the lungs, kidneys, liver, and reproductive tract of both male and female MISTRG mice (Fig 1D). The flow cytometric quantification of DLBCL cell engraftment in the bone marrow of femur and tibia and in jaw bone marrow (in the case of RC-K8) and spleen revealed that a substantial fraction of leukocytes in both tissues were of human (i.e., DLBCL cell) origin, i.e., hCD45- or hCD19-positive, and had also retained their ZsGreen signal during in vivo growth (Fig 1E–G). In the time frame of up to 6 weeks after tumor cell injection assessed here, DLBCL cell engraftment was accompanied by clinical symptoms in only a small fraction (< 20%) of mice; if they occurred, symptoms included weight loss and progressive paralysis of the hind legs, which in some instances could be attributed to tumor growth in close proximity to the spinal cord. In conclusion, MISTRG mice represent a highly permissive host strain for orthotopic DLBCL engraftment that can be monitored over time using IVIS, and that to some extent recapitulates hallmarks of human DLBCL in terms of tissue tropism and aggressiveness.

### Reconstitution of MISTRG mice with cord blood hematopoietic stem cells facilitates DLBCL growth in subcutaneous and orthotopic models

MISTRG mice express several human cytokines and growth factors under the control of their endogenous murine regulatory elements, which not only facilitates DLBCL cell engraftment as shown above, but also permits their reconstitution with a normal human immune system from cord blood-derived hematopoietic stem and progenitor cells (HSPCs; Rongvaux et al, 2014). We speculated that chimeric mice harboring a normal human immune system might support DLBCL engraftment even better than their un-reconstituted counterparts. To assess a possible benefit of a normal human immune system for DLBCL growth, we intrahepatically reconstituted newborn mice with CD34[+] human HSPCs, allowed the human cells to engraft for 6–8 weeks, and subsequently transplanted the mice subcutaneously or orthotopically with U-2932 or RC-K8 cells. The reconstitution efficiency of the mice varied substantially, as assessed at the study endpoint, in both the bone marrow (20–90% human cells among all CD45[+] leukocytes) and the blood (1–20%; Fig 2A). Mice with a reconstitution efficiency below 5% were considered as un-reconstituted. A detailed multi-color flow cytometric analysis of human leukocytes in the bone marrow confirmed that HSPCs in MISTRG mice give rise to the full spectrum of leukocytes that is

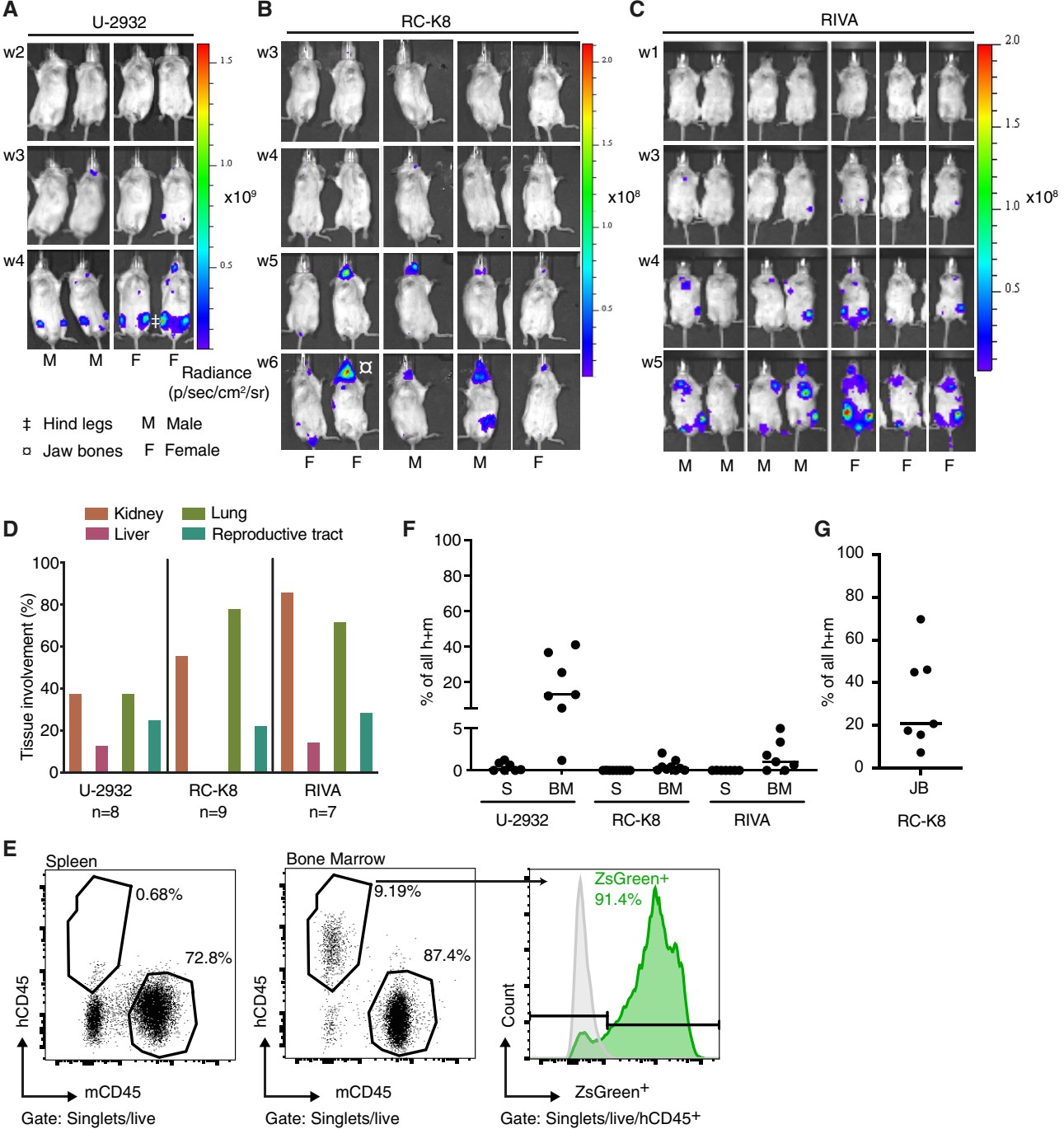

**Figure 1. DLBCL cell lines engraft and form orthotopic lymphomas in MISTRG mice that can be traced by luciferase expression.**

A–C A total of $1 \times 10^7$ ZsGreen- and luciferase-expressing U-2932, RC-K8, and RIVA cells were intravenously injected into 6-week-old male (M) and female (F) MISTRG mice and monitored weekly using IVIS for at least four and up to 3 weeks. The color scales on the right indicate the radiance, i.e., the sum of the photons per second from each pixel inside the ROI/number of pixels (photons/s/cm²/sr).

D The frequency of involvement of the indicated tissues is shown for one cohort of mice and is representative of two independently analyzed cohorts per cell line.

E Gating strategy for the identification of U-2932 cells residing in spleen and bone marrow, among all leukocytes in these organs. Cells were stained for human and mouse CD45. Very few hCD45+ U-2932 cells were detectable in the spleen, whereas their frequencies were much higher in the bone marrow (BM). In BM, > 90% of U-2932 cells were positive for ZsGreen. The green curve shows U-2932 cells recovered from BM, overlayed with unlabeled U-2932 cells grown *in vitro* (gray curve).

F, G Frequencies of human lymphoma cells, as judged by hCD45 expression, among all (human and mouse, h + m) CD45+ leukocytes in the spleen (S), femoral bone marrow (BM), and jaw bone marrow (JB) at the study endpoint of the mice shown in (D) (4 weeks post-injection for U-2932, 5 weeks p.i. for RIVA, and 6 weeks p.i. for RC-K8).

Data information: Data in (A–G) are representative of at least two independently conducted studies per cell line. In the study shown, $n = 7$ (RIVA), 8 (U-2932), and 9 (RC-K8) mice. Horizontal lines indicate medians. Statistics do not apply here.

found in humans (Fig 2B; Rongvaux *et al*, 2014). While CD20$^+$ B cells dominated in the bone marrow, we were also able to readily detect CD3$^+$ T cells, CD33$^+$ myeloid cells, and NKp46$^+$ NK cells (Fig 2B, gating strategy in Fig EV2A). Interestingly, reconstituted mice exhibited larger and heavier tumors over time and at the study endpoint than their un-reconstituted counterparts when transplanted subcutaneously with U-2932 cells (Figs 2C and D, and EV2B), with reconstitution efficiencies correlating directly with the tumor burden at the study endpoint (Fig 2E). We speculated that either human CD4$^+$ T cells or human myeloid cells might be contributing to subcutaneous DLBCL growth in this model; however, neither the depletion of CD4$^+$ T cells nor the neutralization of the myeloid cell-specific chemokine CCL2 with specific antibodies had a discernible effect on tumor growth in the subcutaneous model (Fig EV2C and D).

We next assessed a possible benefit of HSPC humanization on the orthotopic growth of intravenously transplanted U-2932 cells; indeed, we detected higher frequencies of U-2932 cells in the bone marrow of reconstituted relative to un-reconstituted mice, and some reconstituted animals showed involvement of the spleen (Figs 2F and G, and EV2E). The same general trend was observed with RC-K8 cells, which also showed superior engraftment in the bone marrow as a consequence of HSPC humanization, but—in contrast to U-2932 cells—no spleen involvement (Fig 2F and G). The higher tumor burden in the bone marrow of RC-K8-transplanted mice was readily visible also by IVIS (Fig 2H). Reconstituted mice exhibited non-lymphoid tissue involvement more frequently than un-reconstituted mice, with the exception of lungs in the RC-K8 model (Fig 2I), and a substantial fraction of reconstituted mice transplanted with RC-K8 cells developed disease-related clinical symptoms such as weight loss and hind leg paralysis (Fig 2J). The combined results from the two (ectopic and orthotopic) models and the two cell lines indicate that non-malignant human leukocytes can support DLBCL engraftment and growth at lymphoid and non-lymphoid sites.

## Human IL-6 promotes the engraftment and growth of IL-6R$^+$ DLBCL cells

Interleukin-6 (IL-6) is a well-documented growth factor and therapeutic target in plasma cell myeloma, and MISTRG mice that have been modified with an additional knock-in allele encoding human IL-6 (MISTRG6) support multiple myeloma engraftment and growth (Das *et al*, 2016). To examine whether the humanization of IL-6, which lacks species cross-reactivity (Kishimoto, 2005), would provide critical signals necessary for DLBCL growth and survival, we injected luciferase-expressing RC-K8 or U-2932 cells into MISTRG and MISTRG6 mice, compared their growth kinetics over time by IVIS, and assessed human cell engraftment by flow cytometric analysis at the study endpoint. RC-K8 cells engrafted earlier and grew faster in the bones of MISTRG6 mice than in their MISTRG (non-IL-6-expressing) counterparts (Fig 3A); the growth advantage due to IL-6 provision was also evident at the study endpoint in both bone marrow and spleen (Fig 3B and C). Several of the MISTRG6 mice, but none of the MISTRG mice harboring RC-K8 cells, developed disease-related symptoms such as hind leg paralysis; tissue involvement as judged by macroscopic evaluation or *ex vivo* IVIS was observed mostly in MISTRG6 mice (Fig 3D and E). The

differences between the two host strains were not nearly as obvious upon injection of U-2932 cells; there was no discernible difference in the tumor load in the bone marrow or peripheral tissues, although spleens were only colonized in (some) MISTRG6 and not in MISTRG mice (Fig 3A–E). To assess whether differential expression of the IL-6 receptor could possibly explain the differences in terms of IL-6 dependence observed between the two cell lines, we assessed its surface expression by flow cytometry. Indeed, only RC-K8, but not U-2932, cells strongly expressed the IL-6 receptor α-chain CD126 (Fig 3F). To address whether the growth advantage of RC-K8 cells in MISTRG6 mice could indeed be attributed to CD126 expression, we targeted the second exon of the corresponding gene *IL6R* by CRISPR, which resulted in the complete loss of CD126 surface expression (Fig EV3A). RC-K8 cells that had lost CD126 expression colonized MISTRG6 bone marrow somewhat less efficiently than their wild-type counterparts (Fig EV3B). The differential benefit of the two examined DLBCL cell lines from IL-6 provision thus mirrors the benefit conferred by human immune cell reconstitution (Fig 2). Indeed, we are able to detect hIL-6 in the serum of HSPC-reconstituted mice, albeit at lower levels than in serum from MISTRG6 mice (Fig EV3C). Surface expression of the IL-6 receptor CD126 appears to contribute to the differential growth of RC-K8 and U-2932 cells in MISTRG and MISTRG6 mice.

## Reconstitution of MISTRG mice with cord blood hematopoietic stem cells, or knock-in of hIL-6, promotes primary DLBCL engraftment

Primary DLBCL cells are notoriously difficult to engraft in mice; published protocols require surgery and implantation of tumor material under the kidney capsule (Chapuy *et al*, 2016), which creates a model that is technically challenging, not scalable, not orthotopic, and does not lend itself to serial transplantation or drug testing. We speculated that MISTRG or MISTRG6 mice, or MISTRG mice harboring a normal human immune system, might be permissible to primary DLBCL cell engraftment and growth. To this end, we obtained the bone marrow aspirate of a patient with stage IVBE ABC-DLBCL with extranodal manifestations in the liver, an IPI score of 5, a diagnosis of splenomegaly, and approximately 7% infiltration of the bone marrow with tumor cells. All available tumor cells (100,000) were intravenously injected into a MISTRG6 recipient, where they engrafted in the bone marrow and spleen within a time frame of 8 weeks. Spleen cells from this donor mouse were serially transplanted into a cohort of MISTRG and MISTRG6 mice, and engrafted exclusively in the latter strain, where they formed splenic lymphomas with bone marrow involvement that were histologically and immunophenotypically indistinguishable from the patient tumor (Fig 4A–E). None of the MISTRG recipients lacking expression of IL-6 showed evidence of lymphoma engraftment (Fig 4A–E). Interestingly, MISTRG recipients that had been reconstituted at birth with a normal human immune system promoted lymphoma engraftment in the bone marrow as efficiently as mice expressing IL-6 (Fig 4F). Immunophenotyping revealed a clonal, lambda light chain-restricted tumor cell population in all lymphoma-bearing mice (Fig 4D and F). The primary lymphoma cells used here express high levels of the IL-6R α-chain and of the signaling chain gp130 and are positive for the phosphorylated form of the signal

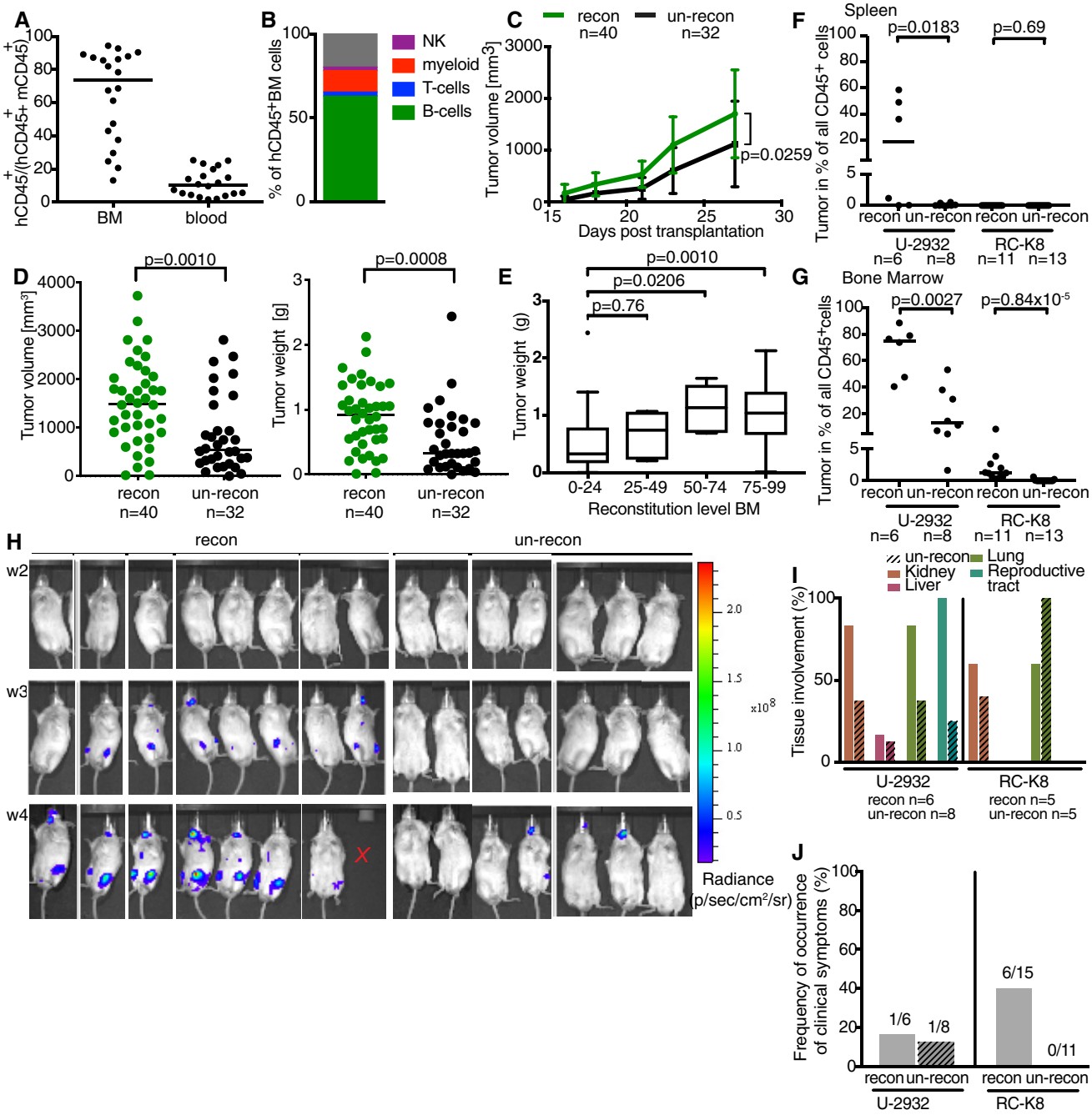

**Figure 2. Reconstitution of MISTRG mice with cord blood hematopoietic stem cells confers a growth advantage to xenotransplanted DLBCL cell lines in ectopic and orthotopic tumor models.**

A–E  MISTRG mice were reconstituted or not at birth with ~ 200,000 human CD34+ cord blood HSPCs and subcutaneously transplanted with 1 × 10^7 U-2932 cells at 6 weeks of age. Tumor volumes were recorded over time, and along with tumor weights, at the study endpoint on day 27 post-injection. The reconstitution efficiency (frequency of human among all human and mouse CD45+ leukocytes) is shown in (A), and the average human leukocyte composition in the bone marrow of the mice in (A) is shown in (B). Tumor volumes over time are presented in (C) along with tumor weights and volumes at the study endpoint in (D), where each data point represents one tumor. The reconstitution level directly correlates with the tumor weight at the endpoint (E).

F–J  MISTRG mice were reconstituted or not at birth with human CD34+ cord blood HSPCs and intravenously injected with 1 × 10^7 U-2932 or RC-K8 cells at 6 weeks of age. The lymphoma burden at the study endpoint was quantified in the spleen (F) and bone marrow (G) by staining for hCD45 and by ZsGreen expression (as well as hCD19 and hCD20 for U-2932 and RC-K8, respectively), and in the case of RC-K8 cells, also over time by IVIS (H). The frequency of tissue involvement was assessed by *ex vivo* IVIS and macroscopic evaluation (I), and the occurrence of clinical symptoms (weight loss, hind leg paralysis) was recorded for each mouse (J). The x indicates that this mouse was not available for IVIS at the final time point because of premature death.

Data information: Data in (A–E) are pooled from three independent experiments. Data in (F–J) are pooled from two independent experiments per cell line. *P*-values were calculated using the Mann–Whitney test and one-way ANOVA (in E). Horizontal lines indicate medians throughout. *n* is indicated in the figure panels throughout. Tumor volumes over time are presented in (C) as mean ± SD. (E) Tumor weights are presented as boxplots with Tukey whiskers for different reconstitution levels in the bone marrow.

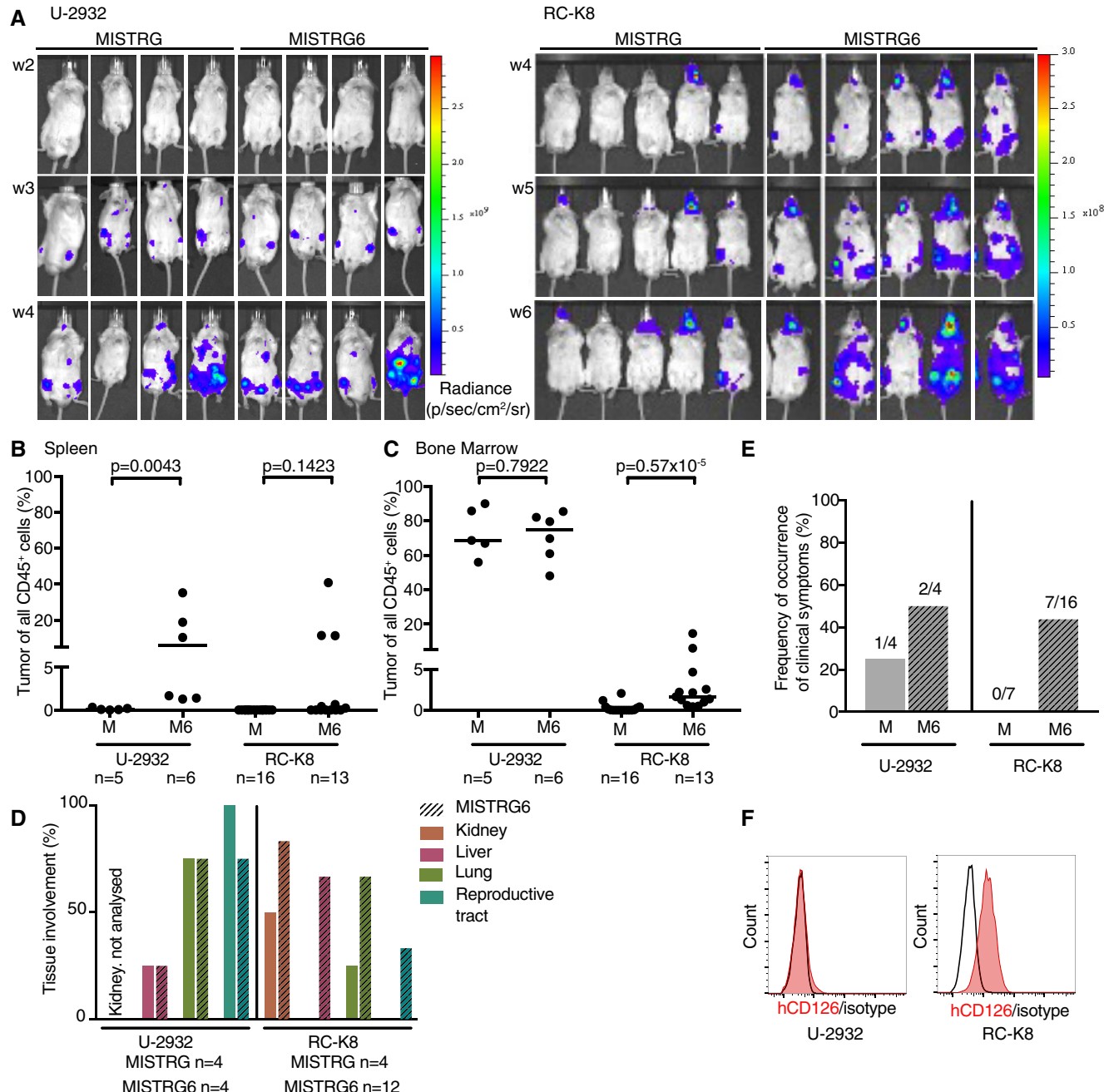

**Figure 3. Genetic IL-6 humanization promotes growth of DLBCL cells that express the IL-6 receptor.**

A–E    MISTRG and MISTRG6 mice were intravenously injected with $1 \times 10^7$ U-2932 or RC-K8 cells at 6 weeks of age and monitored weekly by IVIS until the study endpoint (A). The lymphoma burden at the study endpoint (6 weeks p.i. for RC-K8 and 4 weeks p.i. for U-2932) was quantified in the spleen (B) and bone marrow (C) by flow cytometric staining for hCD45 and by ZsGreen expression. The involvement of the indicated tissues as assessed by *ex vivo* IVIS is shown in (D) (solid bars, MISTRG; hatched bars, MISTRG6), and mice that have developed clinical symptoms (paralysis of hind legs) are plotted in (E). In (B, C, and E), MISTRG is abbreviated as M and MISTRG6 as M6.

F    Surface expression of the IL-6 receptor α-chain (CD126) as assessed by flow cytometry of U-2932 and RC-K8 cells. Data are representative of two stainings.

Data information: Data in (A–E) are from two independent experiments per cell line; *ex vivo* IVIS and clinical scoring of U-2932 (in D and E) was performed for only one experiment, and *in vivo* IVIS is shown for a subset of mice. Horizontal lines indicate medians throughout. *P*-values were calculated using the Mann–Whitney test. *n* is indicated in the figure panels throughout.

transducer and activator of transcription STAT3, which mediates signal transduction downstream of the IL-6R heterodimer (Fig 4G and H). The expression of both chains of the IL-6R could further

also be detected by qRT–PCR of immunomagnetically enriched CD19[+] B cells from this patient's bone marrow aspirate (Fig EV3D and E). To address whether the expression of the IL-6R would

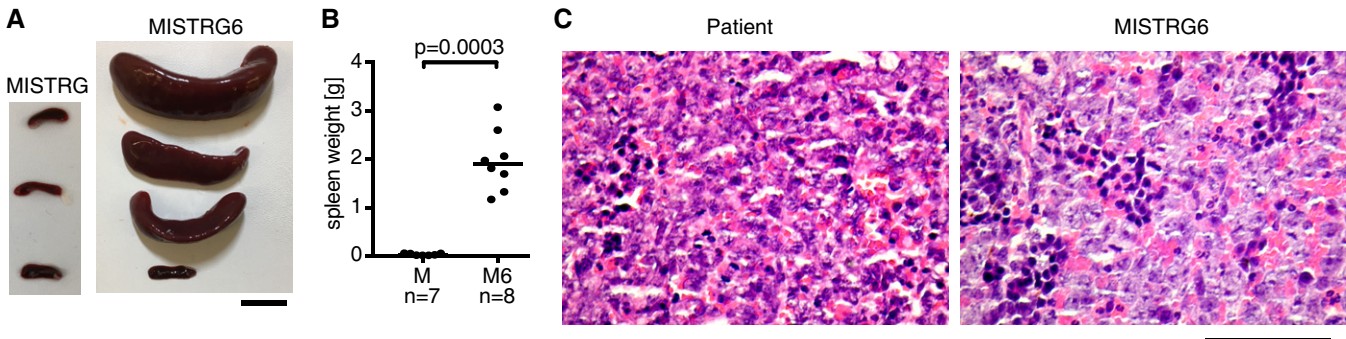

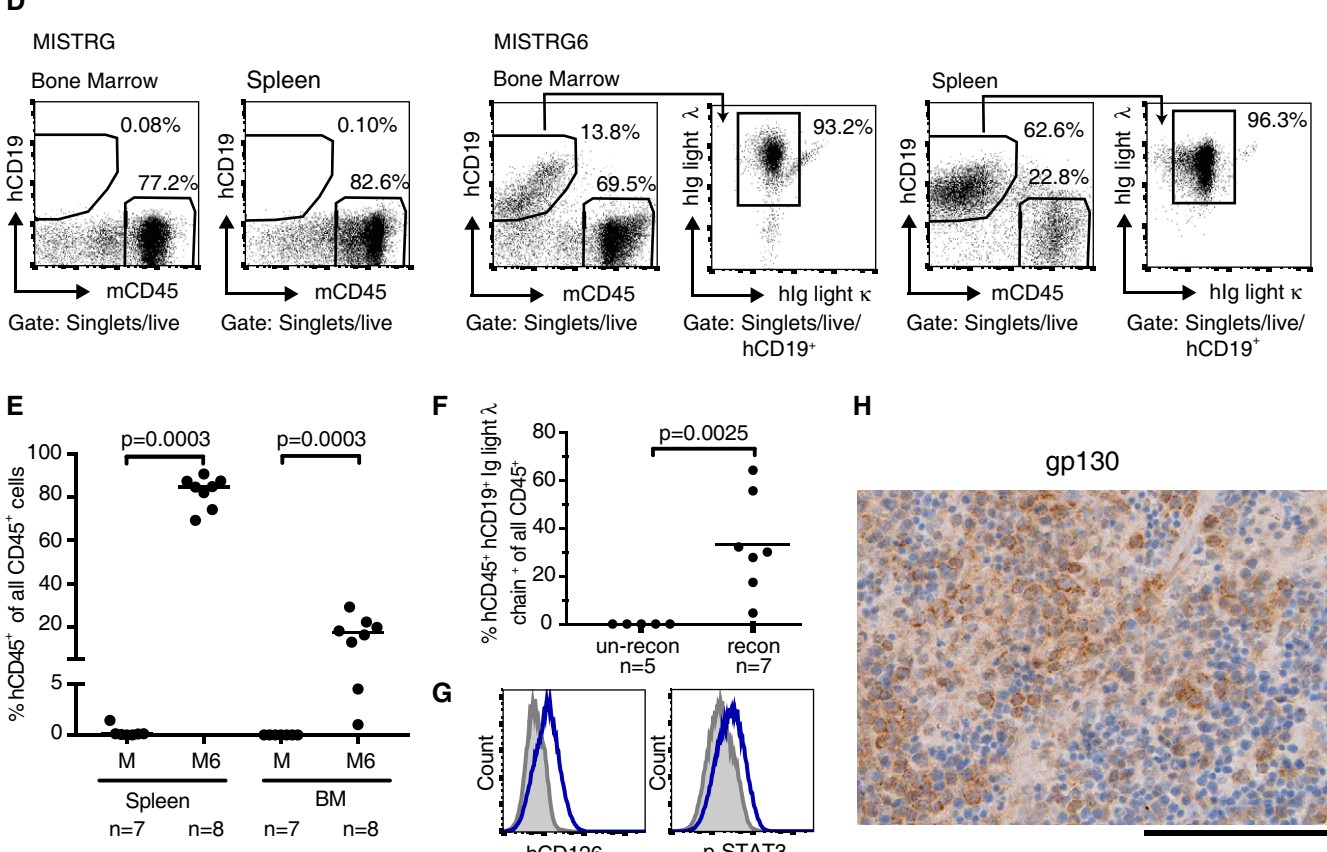

**Figure 4. Human IL-6 and human immune system reconstitution both promote primary orthotopic DLBCL growth.**

A–E   A total of $1 \times 10^6$ primary DLBCL cells obtained by passaging through a MISTRG6 mouse were intravenously injected into MISTRG or MISTRG6 recipients. The spleen weights (B, representative pictures in (A) are shown along with representative H&E-stained spleen (MISTRG6) and lymph node (patient) sections in (C) and the lymphoma burden at the study endpoint 8 weeks p.i., as quantified in the spleen and bone marrow in (D and E). Representative flow cytometry plots are shown in (D) for bone marrow and spleen (MISTRG, MISTRG6). Lymphoma cells were positive for hCD19 and were lambda light chain-restricted (right panels of MISTRG6). M, MISTRG; M6, MISTRG6. Scale bar in (A), 1 cm. Scale bar in (C), 100 μm.

F   MISTRG mice were reconstituted or not at birth with human CD34[+] cord blood HSPCs and intravenously injected with $1 \times 10^6$ primary DLBCL cells at 6 weeks of age. The lymphoma burden at the study endpoint was quantified in the bone marrow of reconstituted and un-reconstituted animals by flow cytometric staining for the indicated surface markers.

G, H   Primary lymphoma cells harvested at the study endpoint express the IL-6R α chain (CD126, as assessed by flow cytometry, G) and the signaling chain gp130 (as assessed by immunohistochemistry, H) and are positive for phosphorylated STAT3 (G). Blue curves, hCD126 or p-STAT3; gray curves, isotype control antibody. Scale bar in (H), 100 μm.

Data information: Data in (A–E) are pooled from two independent experiments and representative of three experiments. Data in (F) are from one experiment and representative of two. In (B, E, and F), each symbol represents one mouse and horizontal lines indicate medians. *P*-values were calculated using the Mann–Whitney test. *n* is indicated in the figure panels throughout.

Source data are available online for this figure.

generally correlate with the propensity for engraftment of primary human DLBCL cells, we subjected three additional patient samples to immunomagnetic enrichment followed by qRT–PCR for both IL-6R chains and transplanted all available cells from each donor (in some cases, this number was under 100,000 cells) into MISTRG6 mice. Of the three additional examined primary cell samples, only one tested positive for both IL-6R chains at the transcript level (Fig EV3D and E). None of the three showed evidence of engraftment in the spleen or bone marrow of MISTRG6 mice in the examined time frame of 8 weeks (Fig EV3E). These results indicate that human IL-6 may promote the orthotopic engraftment and growth of primary DLBCL cells; expression of a functional heterodimeric IL-6R likely is necessary, but not sufficient, for successful engraftment. Once established, primary DLBCL cells can readily be passaged in MISTRG6 mice without losing the characteristics of the human patient material.

## DLBCL cells differ in their IL-6 receptor expression

We next set out to examine more systematically whether the expression of the IL-6R is a common feature of DLBCL cells. In humans, the functional heterodimeric IL-6R consists of the unique subunit CD126 mentioned above and a second subunit, the glycoprotein gp130 (CD130) that is shared with other cytokines including IL-11, IL-27, leukemia inhibitory factor, and oncostatin M, and encodes an intracellular domain required for signal transduction downstream of the functional IL-6R. We assessed the expression of both subunits in a panel of 12 DLBCL cell lines that we had at our disposal. Only two cell lines, the RC-K8 shown above (Fig 3F) and OCI-Ly3, expressed detectable levels of the IL-6R as assessed by flow cytometry and qRT–PCR (Fig 5A and B). The expression of gp130 was difficult to detect by flow cytometry; qRT–PCR results indicate that gp130 is weakly expressed by all ABC-DLBCL cell lines, but not by GCB-DLBCL cell lines (Fig 5A–C). This observation could be confirmed by Western blotting (Fig EV4A). RC-K8 cells express both subunits; this cell line has features of both ABC and GCB subtypes (Kalaitzidis *et al*, 2002; Fig 5C). We next assessed the expression of both chains of the IL-6R in tumor biopsies from a cohort of 114 DLBCL patients that were spotted on tissue microarrays. Staining for the unique IL-6R α-chain proved impossible on formalin-fixed material with commercially available antibodies; however, the expression of gp130 was readily detectable in tumors from 77 of the 114 patients (Fig 5D and E). gp130 expression was more common in ABC-DLBCL cases than in GCB-DLBCL (Fig 5E), with the two subtypes being distinguished by immunohistochemistry based on the Hans classification, which takes CD10, BCL-6, and MUM1 expression into account and is approximately 80% accurate in stratifying the two subsets (Hans *et al*, 2004). In conclusion, a subset of DLBCL cell lines and primary tumors express the IL-6R or its signaling chain, respectively, with a slight bias of expression toward the more aggressive ABC subtype of the disease.

## The constitutive activation of STAT3 is driven by the IL-6/IL-6R axis in a subset of DLBCL

To address whether DLBCL cell lines not only express the IL-6R but possibly also produce IL-6, we performed ELISA on culture supernatants. Only two of the ABC and none of the GCB-DLBCL cell lines produced IL-6 (Fig 6A). Interestingly, the two (ABC subtype) cell lines SU-DHL-2 and OCI-Ly3 that produce IL-6 and express gp130 exhibit constitutive phosphorylation and activation of STAT3 as assessed by Western blotting and flow cytometric analysis of STAT3 phosphorylation on tyrosine 705 (Fig 6B and C). In contrast, all other ABC-DLBCL cell lines (with one exception) that express gp130 activate STAT3 only upon exposure to exogenously added IL-6 (Fig 6B). None of the GCB-DLBCL cell lines (except SU-DHL-5) show constitutive STAT3 activation or respond to IL-6 *in vitro* despite producing STAT3 at similar levels as the ABC-DLBCL cell lines (Fig 6D). RC-K8 cells lacking the IL-6R α-chain CD126 due to genomic editing by CRISPR (Fig EV3A) fail to phosphorylate STAT3 upon addition of IL-6 (Fig 6E). To address whether the constitutive STAT3 activation that is a hallmark of the IL-6-expressing cell lines OCI-Ly3 and SU-DHL-2 is indeed due to IL-6, we treated these cells with an IL-6-specific antibody to neutralize the endogenously expressed cytokine. Only one cell line (OCI-Ly3), but not the other (SU-DHL-2), responded to IL-6 neutralization with a reduction in STAT3 phosphorylation (Fig 6F). Interestingly, SU-DHL-2, but not OCI-Ly3, cells harbor an inactivating mutation in the suppressor of cytokine signaling *SOCS1*; this mutation is known to abrogate the negative regulation of STAT3 signaling by SOCS1 and to render the pathway constitutively active (Juskevicius *et al*, 2018).

To address whether STAT3 phosphorylation in patients correlates with gp130 expression, we evaluated a possible co-expression of gp130 and phospho-STAT3 in our cohort of DLBCL patients. Of the 72 patients whose tumors were clearly positive for gp130, 38 (53%) were also positive for phospho-STAT3; of 42 gp130-negative tumors, only 13 (31%) were positive for phospho-STAT3 (Fig 6G and H; $P = 0.018$). The combined results indicate that DLBCL cells, especially of the ABC subtype, appear to have evolved mechanisms that ensure STAT3 activation, through either the production of IL-6 or the mutational inactivation of *SOCS1*. The co-expression of gp130 and phospho-STAT3 in patient biopsies lends further support to this hypothesis.

## IL-6 signaling contributes to MYC-driven lymphomagenesis in mice

To address in a genetically engineered model of MYC-driven lymphoma whether IL-6 signaling contributes to the lymphoma burden, we crossed C57BL/6 mice expressing Cre recombinase under the activation-induced cytidine deaminase promoter (*AID*-Cre) with mice harboring floxed *Il6ra* alleles, which resulted in a composite strain lacking IL-6 signaling specifically in the germinal center (GC) B-cell compartment after immunization with sheep red blood cells. *AID*-Cre × *Il6ra*^fl/fl mice were additionally crossed with a mouse strain that expresses MYC under the control of the immunoglobulin heavy chain enhancer (*Emu*-MYC) (Adams *et al*, 1985). Immunization of the resulting offspring induced GC formation, *AID*-Cre activity, and *Il6ra* deletion in GC B cells as reported previously (Hashwah *et al*, 2017). In the absence of the *MYC* transgene, none of the mice of this line developed lymphomas. However, the majority of mice harboring *MYC* developed lymphomas in their axillary and inguinal lymph nodes and spleen, which was somewhat (not significantly) delayed by the genetic ablation of both copies of the *Il6ra* (Fig 7A). Heterozygous mice with one functional copy of

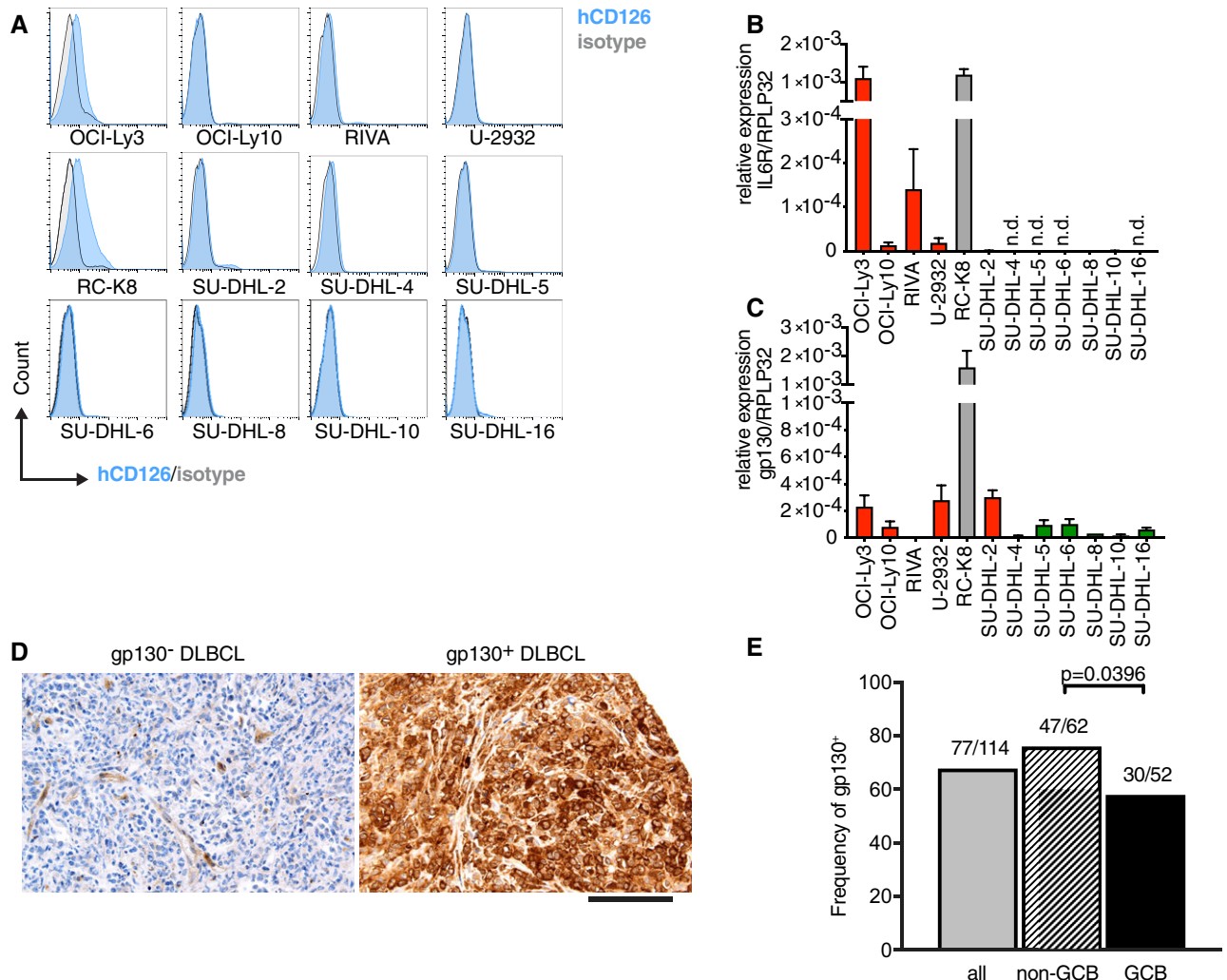

**Figure 5.   The IL-6 receptor is expressed on a subset of DLBCL.**

A–C   The expression of the IL-6Rα (CD126) and signaling chains (gp130) was assessed by flow cytometry (A) and SYBR or TaqMan qRT–PCR (B, C) respectively, on a panel of DLBCL cell lines. ABC- and GCB-DLBCL cell lines are color-coded in red and green; RC-K8 are depicted in gray, as their subtype is a matter of debate (Kalaitzidis *et al*, 2002).

D, E   The expression of gp130 in 114 tumor biopsies of DLBCL patients was assessed by immunohistochemistry and correlated with the ABC/non-GCB vs. GCB subtype, as assessed via the Hans classification. Representative micrographs are shown in (D), alongside the quantification of gp130+ cases among all samples, and among the two subtypes, in (E). Scale bar, 100 μm.

Data information: The FACS plots in (A) are representative of at least two and up to four independent stainings per cell line. qRT–PCR results are presented as mean + SEM of two independent experiments. The *P*-value in (E) was calculated by the chi-square test. n.d., not detectable.

Source data are available online for this figure.

---

*Il6ra* behaved like wild-type mice in this setting (Fig 7A). IL-6ra deficiency in B cells resulted not only in a delay, but also in a reduction of the tumor burden, which manifested in significantly smaller lymph nodes at the study endpoint (Fig 7B). To address whether the hyperproliferation of B cells undergoing the GC reaction upon sheep red blood cell immunization was affected by loss of IL-6ra, we immunized *AID*-Cre × *Il6ra*^fl/fl mice (not harboring the *MYC* transgene) and their heterozygous and wild-type littermates and quantified their GC compartment by flow cytometry and Ki67 immunohistochemistry. The immunization led to a strong increase in CD19⁺CD95⁺CD38^low GC B cells as determined by flow cytometry, which was reduced by the deletion of one, and especially of both alleles of *Il6ra* (Fig 7C). The smaller GC compartment of *AID*-Cre × *Il6ra*^fl/fl mice could be attributed to reduced frequencies of CXCR4^hiCD86^low centroblasts, whereas CXCR4^lowCD86^low centrocytes were unchanged (Fig 7D and E). We further stained spleen sections for Ki67 (Fig 7F) and quantified the GC numbers and area; this approach revealed that the multiplicity of individual GCs as well as their total area was reduced in mice lacking one or both alleles of *Il6ra* (Fig 7G and H). The combined results indicate that IL-6 signaling is GC B-cell-intrinsically required for early-onset, aggressive lymphomagenesis.

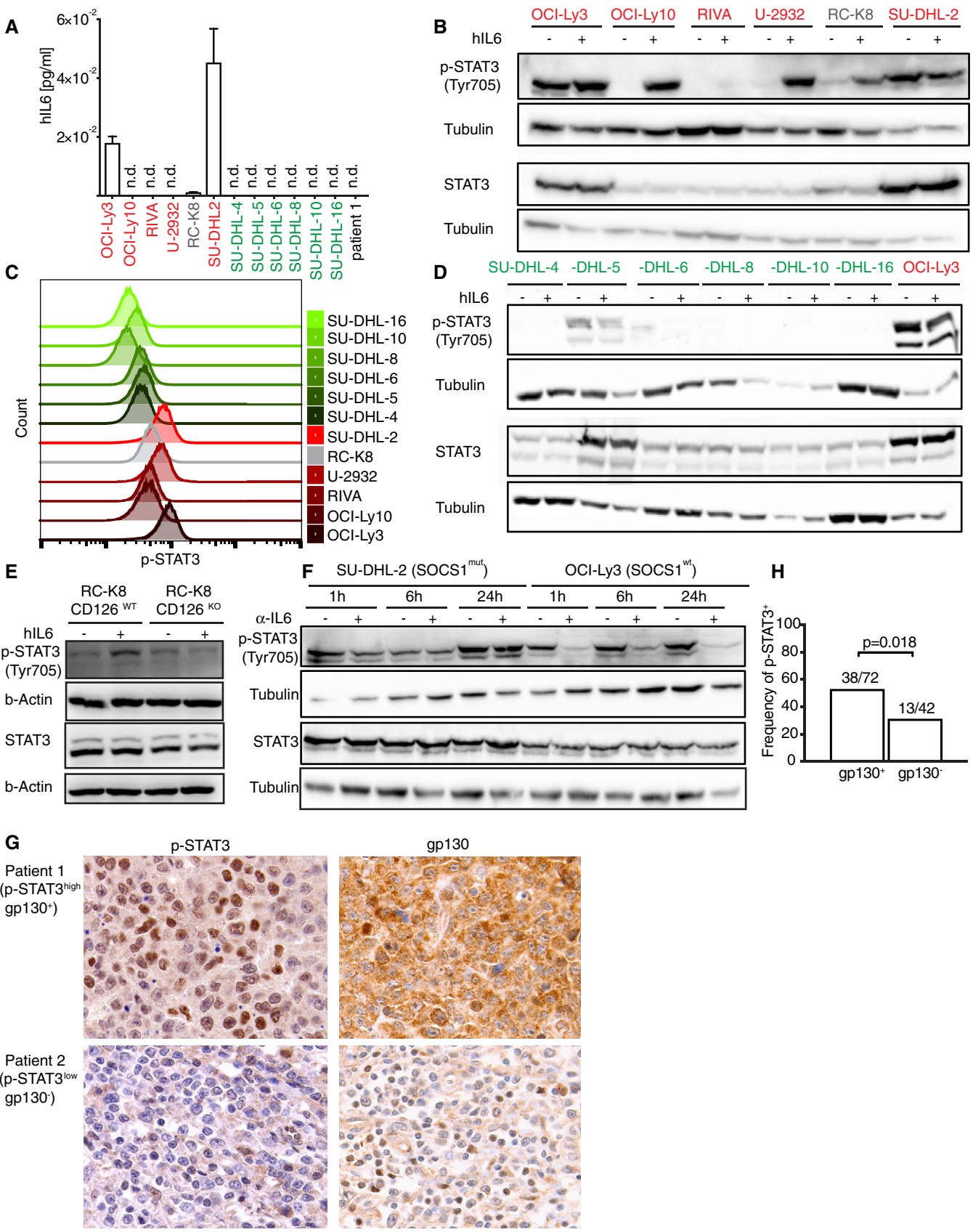

**Figure 6.**

**Figure 6.  ABC-DLBCL cell lines depend on IL-6 for STAT3 activation.**

A  IL-6 secretion of the indicated cell lines (ABC-DLBCL in red, GCB-DLBCL in green) and primary patient material, as quantified by ELISA and normalized to cell number (shown per 1,000 cells). n.d., not detectable.

B–D  STAT3 phosphorylation as assessed by Western blotting (B, D) and flow cytometry (C) of the indicated ABC-DLBCL and GCB-DLBCL cell lines, with and without 1 h of exposure to 50 ng/ml hIL-6 (B, D). STAT3 expression levels are shown as Western blot control. Histograms in (C) represent STAT3 phosphorylation in the absence of exogenously added IL-6.

E  STAT3 phosphorylation as assessed by Western blotting of CD126-proficient (WT) or CD126-deficient (ko) RC-K8 cells, with and without 1 h of exposure to 50 ng/ml hIL-6.

F  Kinetics of STAT3 phosphorylation as assessed by Western blotting, of the IL-6-expressing SOCS1$^{wt}$ and SOCS1$^{mut}$ cell lines OCI-Ly3 and SU-DHL-2, under 1-, 6-, and 24-h exposure to an IL-6-neutralizing antibody (α-IL-6, anti-IL-6-neutralizing antibody).

G, H  STAT3 phosphorylation and gp130 expression of two representative cases (G) and summarized for the entire cohort of 114 patients (H), as assessed by immunohistochemistry. The specific p-STAT3 and gp130 signal is seen as brown (DAB staining), and counterstaining was performed using hematoxylin (blue nuclei). STAT3 phosphorylation is more common in biopsies with gp130 expression; the P-value was calculated by the chi-square test. Scale bar, 100 μm.

Data information: Data in (A) are representative of two experiments and shown as mean + SEM. Data in (B, D, E, and F) were reproduced in three independent experiments.

Source data are available online for this figure.

## The IL-6R/gp130/p-STAT3 axis has negative prognostic impact in both subtypes of DLBCL and can be targeted therapeutically to reduce the tumor burden in orthotopic models

We and others have reported previously that the phosphorylation of STAT3 is a common hallmark of DLBCL patients with unfavorable outcome (Ding *et al*, 2008; Meier *et al*, 2009; Ok *et al*, 2014). We compiled data from these previous reports to specifically address the role of phospho-STAT3 expression in relation to DLBCL subtype and survival probability. Of 814 evaluable patients, 367 were classified as GCB and 376 were classified as non-GCB/ABC-DLBCL according to the Hans algorithm (Hans *et al*, 2004), the subtype being unknown for 71 patients, who were excluded from subsequent analyses. Phospho-STAT3 positivity, defined as expression in ≥ 17% of tumor cells, was not exclusively, but much more commonly found in ABC-DLBCL tumors, and in more advanced (stage 3 and 4, relative to stage 1 and 2) disease (Fig 8A). p-STAT3 positivity was a strong negative prognostic factor in the entire cohort (Fig 8B) and associated with higher overall as well as disease-specific mortality in both ABC- and GCB-DLBCL subtypes (Fig 8C and D). To address whether active STAT3 signaling correlated with activity of the JAK2 kinase, which has previously been implicated in STAT3 phosphorylation in DLBCL (Meier *et al*, 2009), we selected 330 cases of which roughly one half was p-STAT3-positive and the other was p-STAT3-negative. All cases were evaluable by immunohistochemistry for activated (phosphorylated at Tyr1007/1008) JAK2 (Fig EV5A). Regression analysis revealed a positive association between p-STAT3 and pJAK2 (Fig EV5B). Of p-STAT3-positive cases, the vast majority was also positive for pJAK2; however, we also detected a large number of pJAK2-positive cases that were negative for p-STAT3, indicating that additional factors (STAT3 expression or subcellular localization, for example) may affect STAT3 phosphorylation (Fig EV5C). The combined results indicate that the IL-6R/STAT3 signaling axis is preferentially but not exclusively active in ABC-DLBCL, and a negative prognosticator for both subtypes; STAT3 phosphorylation correlates with activation of the upstream kinase JAK2.

IL-6R is a therapeutic target in rheumatoid arthritis (RA); a neutralizing antibody targeting the IL-6R, tocilizumab (trade name: Actemra), is approved for the treatment of RA in adult patients (Ogata *et al*, 2012, 2018) and has also recently received FDA approval for the treatment of cytokine release syndrome related to CAR T-cell therapy (Le *et al*, 2018). We sought to examine a possible benefit of tocilizumab monotherapy in MISTRG6 mice that harbor spleen- and bone marrow-colonizing lymphomas due to intravenous transplantation of serially passaged primary DLBCL cells; this patient-derived xenograft is positive for the IL-6R and also positive for phosphorylated STAT3 (Fig 4). Mice received either an isotype control antibody or tocilizumab starting 2 weeks after lymphoma cell transplantation. Mice on tocilizumab showed a reduction of the spleen size at the study endpoint (Fig 8E); the phosphorylation of STAT3, assessed immunohistochemically on paraffin-embedded material, was reduced in response to the treatment (Fig 8F). We next examined the effects of tocilizumab on MISTRG mice transplanted with the cell line OCI-Ly3, which produces IL-6 (Fig 6A). Indeed, (non-IL-6-expressing) MISTRG mice harboring OCI-Ly3 cells exhibited detectable levels of hIL-6 in their serum (Fig EV5D). Tocilizumab treatment reduced the OCI-Ly3 tumor burden in the bone marrow (Fig 8G). In a third experimental setting, MISTRG6 mice transplanted with RC-K8 cells were subjected to tocilizumab treatment, which also detectably reduced the tumor burden (Fig 8H). The combined results from our patient cohort and xenotransplantation models indicate that active IL-6R/STAT3 signaling is a negative prognostic factor and therapeutic target in DLBCL; biomarkers that may be useful in predicting therapy success include the expression of IL-6R, gp130, and phospho-STAT3.

## Discussion

With this study, we demonstrate that IL-6 signaling is active in a particularly aggressive subset of DLBCL and supports DLBCL growth in relevant *in vivo* models. Before being renamed in 1989 (Kishimoto, 1989), IL-6 was referred to as B-cell stimulatory factor 2 (BSF-2) due to its ability to induce the differentiation of activated B cells into antibody-producing plasma cells (Kishimoto, 1985). While originally described as a growth and differentiation factor for normal B cells, the ability of IL-6 to also stimulate the proliferation of plasma cell-derived plasmacytoma cell lines (Van Snick *et al*, 1987) and primary plasma cell myeloma cells (Anderson *et al*, 1989) was discovered shortly thereafter. In contrast, other malignancies

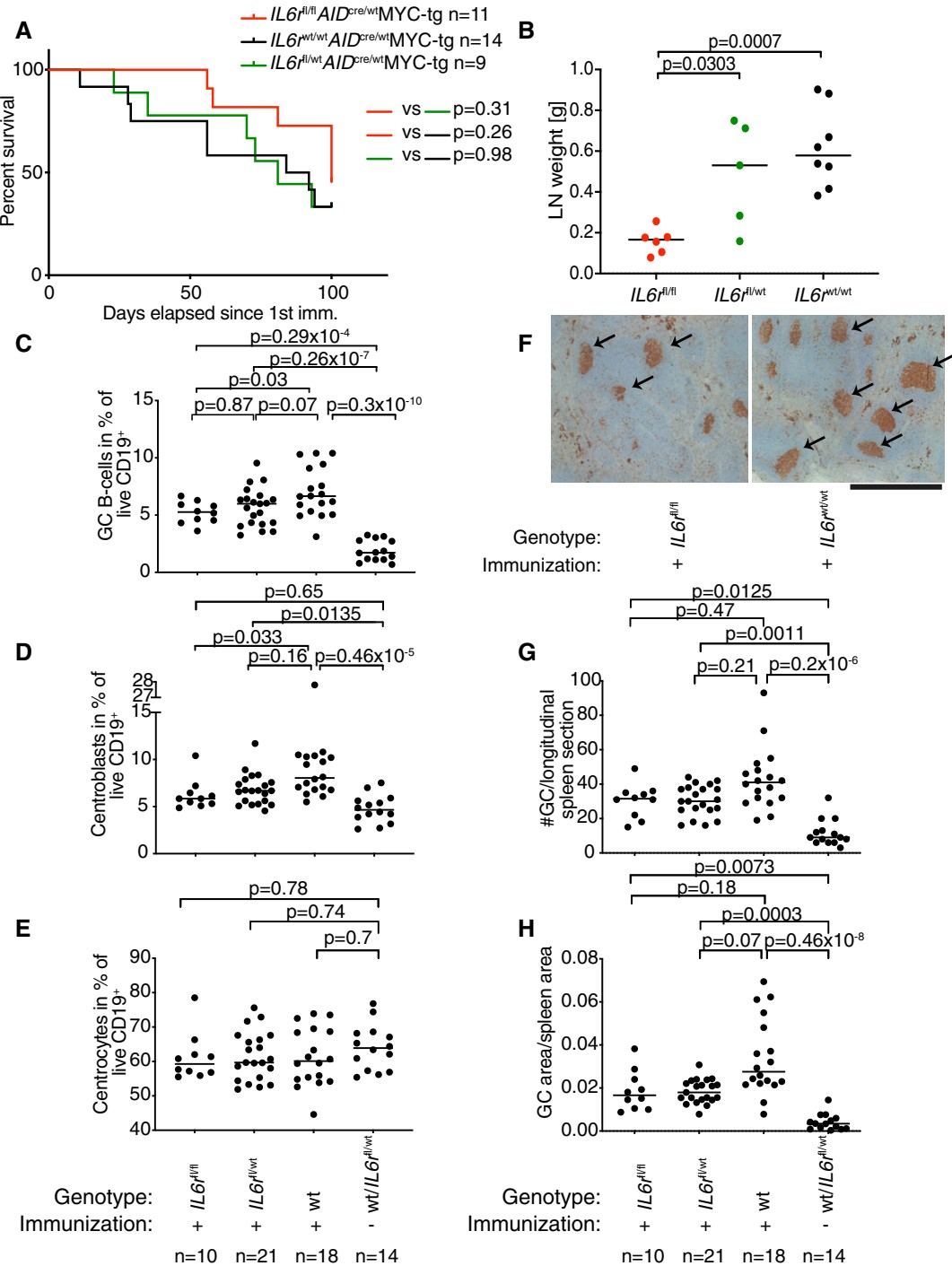

**Figure 7. IL-6 signaling contributes to lymphoma development and germinal center B-cell proliferation in a genetically engineered mouse model.**

A, B  Disease-specific mortality (A) and lymphoma burden as determined by lymph node weight (B), of cohorts of 9–12 mice each (*n* indicated in the figure) of the indicated genotypes.

C–H  Mice of the indicated genotypes were immunized once with sheep red blood cells and assessed 10 days later by flow cytometry of splenocytes for their frequencies of GC B cells (C), centroblasts (D), and centrocytes (E). Spleen sections were further stained for Ki67 to reveal GCs (F), which were quantified in both number (G) and area (H). Arrows in (F) point to GCs; scale bar, 1,000 μm.

Data information: Data in (A and B) are pooled from three (A) and two (B) different litters. Data in (C–H) are pooled from three independent studies. Horizontal lines indicate medians throughout. Statistical comparisons in (B–H) were performed either by one-way ANOVA (in the case of normal data distribution) or by non-parametric ANOVA (Kruskal–Wallis test, in the case of non-normal data distribution) with Tukey's multiple comparisons correction. *n* is indicated in the figure panels throughout.

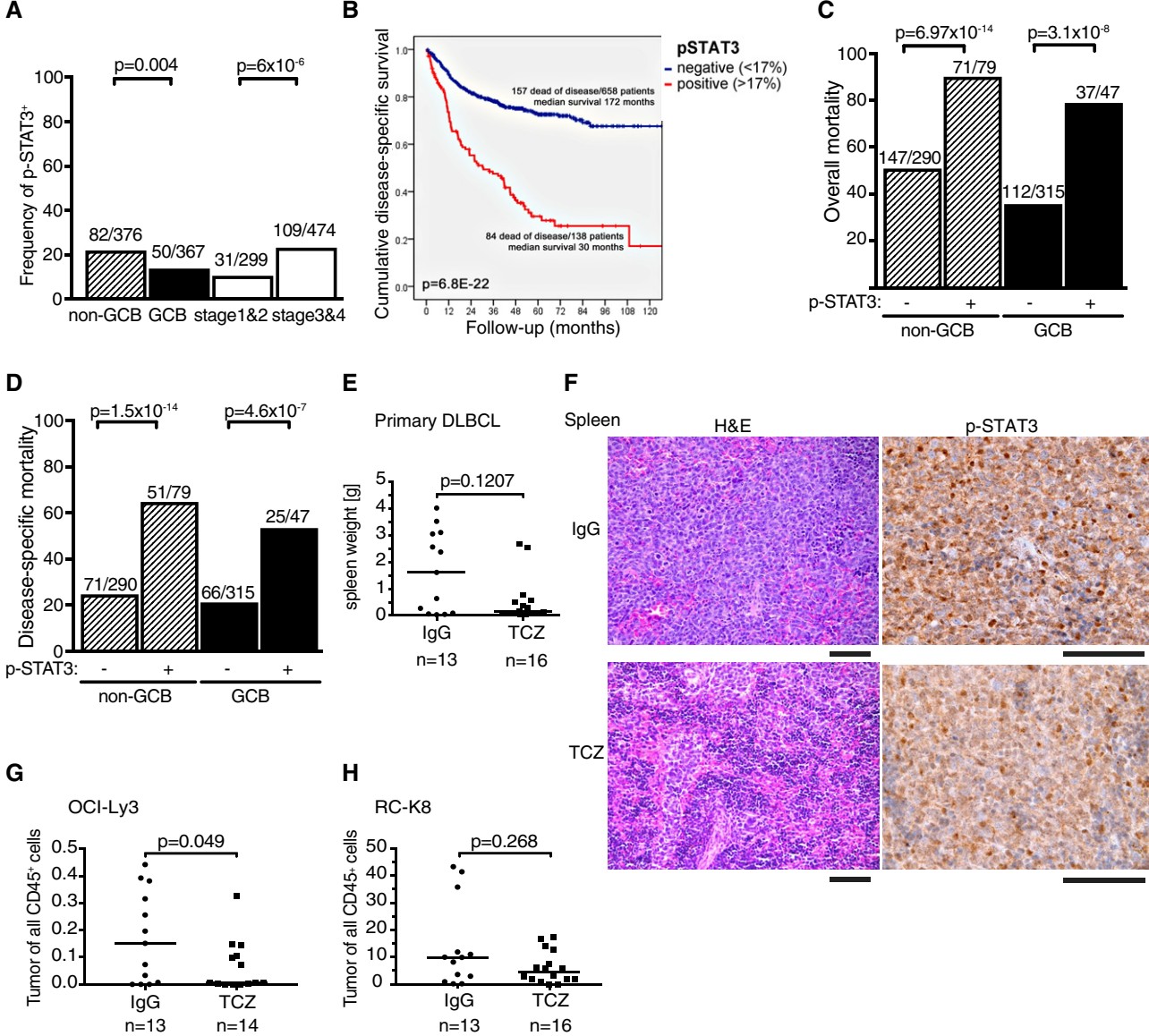

**Figure 8. The IL-6 signaling complex is of diagnostic relevance and a therapeutic target in IL-6R⁺ DLBCL.**

A–D  Frequency of STAT3 phosphorylation (A), and overall as well as disease-specific survival probability (B), and of overall and disease-specific mortality of DLBCL patients stratified according to subtype and disease stage (C and D). Of all 814 evaluable patients, 367 were classified as GCB and 376 were classified as non-GCB/ABC-DLBCL according to the Hans algorithm. Phospho-STAT3 positivity was defined as expression in ≥ 17% of tumor cells and more commonly found in ABC-DLBCL tumors, and in more advanced (stage 3 and 4, relative to stage 1 and 2) disease (A). p-STAT3 positivity was a strong negative prognostic factor in the entire cohort (B) and associated with higher overall as well as disease-specific mortality in both ABC- and GCB-DLBCL subtypes (C and D).

E–H  Effect of IL-6R neutralization by twice-weekly injection of 250 μg/dose of tocilizumab or an isotype control antibody, starting 2 weeks after tumor cell injection, on (E and F) the spleen weights of MISTRG6 mice transplanted with $1 \times 10^6$ primary DLBCL cells (expanded by one serial passage through MISTRG6 mice), on the bone marrow lymphoma burden of OCI-Ly3-transplanted MISTRG mice (G), and on the splenic lymphoma burden of RC-K8-transplanted MISTRG6 mice (H), all assessed at around 4 (G and H) to 8 (E) weeks after tumor cell injection. Representative microscopy images in (F) show STAT3 phosphorylation in spleen-colonizing primary cells exposed to either tocilizumab or isotype control antibody (in brown, counterstained in blue with hematoxylin), and an H&E-stained section from the same mouse. TCZ, tocilizumab. Scale bars, 100 μm.

Data information: Results in (E, G, and H) are pooled from two (E and G) or three (H) independent studies each. Horizontal lines indicate medians. *P*-values were calculated by the chi-square test (A, C, D), by log-rank test (B), or by the Mann–Whitney test (E, G, H). *n* is indicated in the figure panels throughout. Source data are available online for this figure.

---

derived from the B-cell lineage were not previously known to respond to IL-6. IL-6 signals through its unique ligand-binding IL-6Rα chain, expressed mainly by hematopoietic cells and hepatocytes, and the common signal-transducing chain gp130, which is much more ubiquitously expressed on most cells of the body and is also engaged by receptors specific for IL-11, IL-27,

leukemia inhibitory factor, oncostatin M (OSM), ciliary neurotrophic factor (CNTF), and cardiotrophin 1 (CT1). The signaling pathway is further complicated by the fact that a soluble form of the IL-6Rα, generated through proteolytic cleavage by the metalloproteinase ADAM10 or ADAM17, may also bind to IL-6 and induce signaling in cells that do not express the membrane-bound form of IL-6Rα. Upon binding of the (soluble or membrane-bound) IL-6/IL-6Rα complex to gp130, a gp130-associated Janus kinase (JAK; preferentially JAK1/2) activates STAT3 through phosphorylation of its tyrosine residue 705. We report here that DLBCL cells, especially of the ABC subtype, have evolved several complementary mechanisms that ensure constitutively active IL-6 signaling. All ABC-DLBCL, but none of the investigated GCB-DLBCL, cell lines express gp130 at least at low levels; the expression of gp130 appears to be a prerequisite for the ability of the cell lines to activate STAT3 upon exposure to IL-6, as none of the GCB cell lines showed STAT3 phosphorylation upon IL-6 treatment, despite expressing copious amounts of STAT3 itself. Two-thirds of a patient cohort of 114 patients were positive for gp130 as determined by immunohistochemistry, again with a clear bias of expression toward the ABC-DLBCL subtype. A second adaptation of ABC-DLBCL that is consistent with the putative dependence of this subtype on IL-6 is the production and secretion of the cytokine itself. The main cellular source of IL-6 in the steady state and in settings of infection, inflammation, and carcinogenesis is now widely believed to be of myeloid origin; however, normal B cells and plasma cell myeloma cells may also produce IL-6 and bind it in an autocrine manner. Our in vitro experiments clearly revealed that only IL-6-expressing cell lines show constitutive STAT3 activation, which in turn is abrogated by neutralization of IL-6. Again, the analysis of gp130 and phospho-STAT3 co-expression in our patient cohort confirmed that active STAT3 is more commonly (but not exclusively) observed in gp130-expressing DLBCL. As STAT3 signals not only downstream of gp130, but also downstream of the receptors for IL-10, type I interferons, and EGF, it is conceivable and even likely that alternative microenvironmental signals may contribute to STAT3 activation in the context of DLBCL. A third adaptation of ABC-DLBCL cells attesting to their dependence on constitutive STAT3 activation is the mutational inactivation of *SOCS1*. SOCS1 negatively regulates STAT3 phosphorylation downstream of gp130; indeed, the only ABC-DLBCL cell line in our panel that harbors an inactivating *SOCS1* mutation (Juskevicius *et al*, 2018) was independent of IL-6 with respect to its constitutive STAT3 activation. *SOCS1* mutations are rather common in DLBCL; in a patient cohort of 138 primary untreated DLBCL patients that we subjected to targeted high-throughput sequencing of either all exons or the hotspots of 68 frequently mutated genes in B-cell lymphomas, *SOCS1* mutations emerged as the second most common mutation, affecting 28% of patients (Juskevicius *et al*, 2017). Forty-seven different mutations, the vast majority predicted to be deleterious for protein synthesis or function, of *SOCS1* were detected in 21 patients, of which 12 were classified as non-GCB and 9 as GCB-DLBCL (Juskevicius *et al*, 2017). Other studies have reported similar frequencies, with 16 (Schif *et al*, 2013) to 27% (Mottok *et al*, 2009) of patients harboring *SOCS1* mutations.

As a consequence of their dependence on IL-6, DLBCL cell lines and primary cells expressing the IL-6R benefit from a microenvironment that provides access to the cytokine. MISTRG mice expressing human IL-6 from an allele knocked into the endogenous murine locus, and MISTRG mice reconstituted with a normal human immune system, have emerged in this study as terrific hosts for DLBCL engraftment, with involvement of the bone marrow, spleen, and various internal organs. Cell lines that express the IL-6Rα (RC-K8) colonize primary and secondary lymphoid tissues of MISTRG6 mice at much higher levels than those of MISTRG mice; the same is true for primary cells, which do not engraft at all in the absence of IL-6. We engineered our cell lines to express luciferase, which allowed us to track them, and to quantify the tumor burden over time. The examined DLBCL cell lines all showed unique and highly reproducible tissue tropisms that in some instances may be reflective of the parental tumors. RC-K8 cells, for example, initially grow mostly in the jaw bone, and only later, and in the presence of IL-6, will disseminate to other bones and organs. The orthotopic growth of the cell lines in both lymphoid and non-lymphoid organs is reflective of the heterogeneity of human DLBCL, which may arise at, or disseminate to, many different sites, including the gastrointestinal tract, reproductive organs, and brain. Due to its versatility, ease of use not requiring surgery, rapid disease course, and traceability by IVIS, this novel model lends itself to a variety of applications, including (co-clinical) drug susceptibility testing. We have used the model to deliver proof of concept that targeting the IL-6 signaling pathway is a promising strategy for the treatment of IL-6R$^+$/phospho-STAT3$^+$ cases of DLBCL. The antibody we used for this purpose, tocilizumab, is a humanized anti-IL-6Rα monoclonal Ab of the IgG1 class that was generated by grafting the complementarity-determining regions of a mouse anti-human IL-6R Ab onto human IgG1 (Sato *et al*, 1993). It blocks IL-6-mediated signal transduction by inhibiting IL-6 binding to the transmembrane and soluble forms of IL-6Rα. The efficacy, excellent tolerability, and safety of tocilizumab have been verified in numerous clinical trials resulting in the approval of this biologic for the treatment of RA (Tanaka *et al*, 2014), juvenile idiopathic arthritis (Yokota *et al*, 2008, 2012), and Castleman's disease (Nishimoto *et al*, 2008), the latter being a non-cancerous disorder of the lymph nodes characterized by non-clonal hyperproliferation of B cells resulting from IL-6 hypersecretion. The latest indication for which tocilizumab has been approved is severe cytokine release syndrome due to CAR T-cell therapy (Teachey *et al*, 2013; Le *et al*, 2018). Although the early findings describing the IL-6 dependence of plasma cell myeloma cells had generated a strong interest in the IL-6 signaling complex as a therapeutic target in this disease, the initial clinical trials were disappointing; a phase II trial in high-risk smoldering myeloma patients is ongoing (Matthes *et al*, 2016).

In conclusion, we show here that a sizable fraction of DLBCL patients, especially of the ABC subtype, exhibit hallmarks of aberrant activation of the IL-6 signaling pathway which constitutes a promising therapeutic target in this population. Tumors with putative susceptibility to IL-6 inhibition can be identified based on their IL-6R expression and phospho-STAT3 positivity, which is readily detectable by flow cytometry or immunohistochemistry and is associated with a dismal prognosis. *SOCS1* mutations anywhere in the only exon of the gene would counter-indicate treatment success. In summary, we propose here a novel driver of DLBCL growth that can be therapeutically targeted with a safe and well-tolerated biologic response modifier that is approved for a variety of unrelated indications in humans.

## Materials and Methods

### Cell culture experimentation and ELISA

The DLBCL cell lines used here included six of GCB-DLBCL subtype (SU-DHL-4, SU-DHL-5, SU-DHL-6, SU-DHL-8, SU-DHL-10, and SU-DHL-16) and five of ABC-DLBCL subtype (U-2932, OCI-Ly3, OCI-Ly10, SU-DHL2, and RIVA) and one unclassified cell line (RC-K8) that have all been described previously (Hashwah *et al*, 2017; Stelling *et al*, 2018) and are routinely tested, and negative, for mycoplasma. Cell line authentication was performed according to the guidelines of the International Cell Line Authentication Committee using short tandem repeat profiling; the result of this effort was described recently for all cell lines used here, along with their sources (Juskevicius *et al*, 2018). Cell lines were maintained at 37°C and 5% $CO_2$ in a humidified atmosphere in IMDM (RIVA, OCI-Ly3, OCI-Ly10) or RPMI (SU-DHL2 and SU-DHL5) supplemented with 20% (SU-DHL-4, SU-DHL-6, SU-DHL-10, SU-DHL-16, RC-K8, U-2932) or 10% (all others) heat-inactivated FBS and antibiotics. Cells in culture were treated with 50 ng/ml of human recombinant IL-6 (Stemcell Technologies), and supernatants were collected 1 h after treatment and used for protein extraction. For neutralization of human IL-6 bioactivity, anti-human IL-6 antibody (MQ2-13A5) or Rat IgG1κ Isotype Ctrl (RTK2071) antibodies (Biolegend) were used at final concentrations of 8 μg/ml. The levels of secreted IL-6 in culture supernatants and in mouse serum were determined by assaying supernatants collected from cells seeded at 400,000/ml 2 days earlier or from serum harvested at euthanasia with the Human IL-6 ELISA Kit (Thermo).

### Lentiviral gene transfer and genomic editing

A lentiviral approach was used to generate luciferase-expressing DLBCL cell lines. Lentiviral packaging was performed in HEK 293T cells (cultured in DMEM (Gibco) supplemented with 10% FBS and 100 U/ml penicillin and 100 μg/ml streptomycin) by polyethylenimine transfection (PEI, Polysciences, MW 25000). In brief, for each transfection, 4 μg of viral plasmid pHIV-Luc-ZsGreen (Addgene plasmid #39196), 2 μg of helper plasmid psPAX2 (Addgene plasmid #12260), and 1 μg of envelope plasmid pCMV-VSV-G (Addgene plasmid #8454) were used. DLBCL cell lines were infected with lentiviral particles by spinoculation at 32°C and $800 \times g$ for 60 min in the presence of 5 μg/ml polybrene (hexadimethrine bromide; Sigma-Aldrich). Transduced cells (high ZsGreen-expressing) were sorted using a FACSAria III at day 6 after transduction. CRISPR/Cas9 genomic editing of the *IL6RA* gene (encoding the IL-6R α-chain CD126) in RC-K8 cells was achieved by electroporation. The PX458 plasmid used was obtained from Addgene (#48138: pSpCas9 (BB)-2A-GFP). The *IL6RA* guide (sequence: GTCGGTGCAGCTCCACGACTC) targeting the second exon of the gene was cloned into the PX458 vector. Nucleoporation of $1 \times 10^6$ RC-K8 cells was performed with 0.5–3 μg DNA using the Amaxa Nucleofector II device. At 48 h post-transfection, cells were sorted based on their GFP expression; this first sort was followed by two subsequent sorts (at 1 and 2 weeks post-transfection) for $CD126^+$ and $CD126^-$ fractions, which were tested side by side in all experiments. Sorted bulk cells were subjected to flow cytometric

verification of the CD126 knockout using a specific antibody (clone UV4; Biolegend).

### Animal experimentation, reconstitution, *in vivo* imaging, and tissue processing and staining

M-CSF^h;IL-3/GM-CSF^h;hSIRPA^tg;TPO^h;Rag2^−;γc^− (MISTRG; Rongvaux *et al*, 2014) and MISTRG mice additionally expressing IL-6^h (MISTRG6; Das *et al*, 2016) were obtained from a local repository. For xenotransplantation studies, cell lines were injected subcutaneously ($1 \times 10^7$ cells in 150 μl PBS) into both flanks of 6- to 8-week-old mixed-gender MISTRG mice, or intravenously ($1 \times 10^7$ cells in 100 μl PBS) into mixed-gender MISTRG or MISTRG6 mice in the orthotopic model. Once palpable tumors had formed in the subcutaneous model ($\sim 40$ mm$^3$), the volume of the tumors was measured by calipers and calculated using the formula $(a^2 \times b)/2$, where $a$ is the shorter and $b$ the longer tumor dimension. For whole-body bioluminescent imaging of mice included in the orthotopic model, mice were i.p. injected with D-luciferin (150 mg/kg; VivoGlo Luciferin; Promega) and mice were analyzed 10 min later using an IVIS Spectrum system (Caliper LifeSciences). Bioluminescence images were acquired using "auto" setting with F/stop = 1 and binning = medium. A digital false-color photon emission image of the mouse was generated, and photons were counted within the whole-body area. Photon emission was measured as radiance in p/s/cm$^2$/sr. At the study endpoint of orthotopic models, spleens were harvested and weighed, and hind legs/jaw bones were collected for the analysis of the bone marrow. Tissue colonization by lymphoma cells was assessed for the liver, kidney, lung, and reproductive organs after removal from the carcass using two complementary approaches: Livers and kidneys were first macroscopically inspected for the appearance of white nodules indicative of tumor growth, and then placed individually, alongside the lung lobes and ovaries or testicles into 12-well plates filled with luciferin in PBS (300 μg/ml) for *ex vivo* IVIS imaging. Organs were imaged within 5 min of euthanasia. For the reconstitution of mice with a human immune system, 2- to 4-day-old pups were irradiated with a sublethal dose of 1 Gray and intrahepatically injected with 100,000–250,000 immunomagnetically isolated $CD34^+$ cord blood-derived hematopoietic stem cells as described previously (Arnold *et al*, 2017). Reconstituted mice were included in experiments at 6 weeks of age. Anti-human CD4 (OKT-4), mouse IgG2b isotype control, anti-mouse/human/rat CCL2 (2H5), and polyclonal Armenian hamster IgG antibodies were purchased from BioXCell. Mice were administered a 500 μg dose 1 day before subcutaneous injection of tumor cells, then once or twice per week with a 250 μg of anti-human CD4 or anti-CCL2, respectively, until the end of the study (d27).

For patient-derived xenograft transplantation, bone marrow mononuclear cells (MNCs) containing DLBCL cells (obtained from Clinic of Hematology-Oncology at the University of Zürich) were intravenously injected into MISTRG6 mice. Lymphoma cells engrafted in the spleen were serially transplanted ($1–2 \times 10^6$ cells) to MISTRG and MISTRG6 mice, and 8 weeks after tumor cell injection, bone marrow of hind legs and spleens were harvested and analyzed by flow cytometry. Treatment with recombinant anti-human CD126 antibody (Actemra®, Roche) and recombinant human IgG1 isotype control antibody (BioXCell, BE0297) was performed

two times per week. During the first week of treatment for each injection, 500 μg of the respective antibody was administered i.p. During the remaining study period, mice were injected with 250 μg antibody per dose. Serially transplanted patient material was enriched for human B cells using the MagniSort™ Human CD19 Positive Selection Kit (Thermo). Cells were then either used for RNA extraction or cultured for 2 days in RPMI 1640 (supplemented with 20% FBS, 1 mM sodium pyruvate, non-essential amino acids, 100 U/ml penicillin, 100 μg/ml streptomycin) for ELISA. For qRT–PCR-based determination of gp130 and IL-6RA expression, bone marrow MNCs containing DLBCL cells were enriched for human B cells as described above and then subjected to RNA extraction and qRT–PCR (see below).

The mouse strains B6;SJL-$Il6ra^{tm1.1Drew}$/J, B6.Cg-Tg(IghMyc)22Bri/J, and B6;129P2-Aicdtm1CreMnz/J were obtained from the Jackson Laboratories. $Il6ra^{fl/fl}$ mice were crossed to $AID^{cre}$ animals and MYC-transgenic mice to obtain composite strains. For longitudinal monitoring of MYC-transgenic mice, 6- to 8-week-old mice were immunized at regular 14-d intervals with an i.p injection of 200 μl of 10% sheep red blood cells (Innovative Research) and monitored for clinical symptoms and enlarged lymph nodes indicating lymphoma development within a period of 100 days. Moribund mice were euthanized, and their spleens as well as inguinal and axillary lymph nodes were weighed. For immunization studies, mice received a single dose of 100 μl of 4% sheep red blood cells intravenously and were analyzed for germinal center formation 10 days later. Splenocytes were treated with ACK red blood cell lysis buffer pH 7.2–7.4 (150 mM $NH_4Cl$, 10 mM $KHCO_3$, 0.1 mM $Na_2EDTA$) and passed through a 40-μM cell strainer to produce single-cell suspensions. Cells were subsequently stained using the following fluorescent-labeled anti-mouse antibodies: PE-Cy7-conjugated anti-CD95/FAS (Jo2; Becton Dickinson), PerCP-Cy5.5-conjugated anti-CD86 (GL-1; Biolegend), Fixable Viability Dye eFluor® 780, APC-conjugated anti-CD19 (eBio1D3), FITC-conjugated anti-CD38 (90), and PE-conjugated anti-CXCR4 (2B11), all from eBio/Affymetrix. Data were acquired on CyAn ADP (Beckman Coulter) or LSRFortessa (BD) flow cytometers and analyzed with the FlowJo software package (TreeStar). For Ki67 staining of germinal centers, CONFIRM anti-Ki-67 (30-9) Rabbit Monoclonal Primary Antibody (Roche) was used for staining of proliferating cells within germinal centers on formalin-fixed paraffin-embedded spleen sections using a Ventana automated slide stainer. One longitudinal section was assessed per mouse spleen. The images were acquired on a Leica microscope, and ImageJ was used for area assessment. Germinal centers were identified as Ki67-positive areas within blue areas. All animal studies were reviewed and approved by the Zürich Cantonal Veterinary Office (licenses 227/2015, 235/2015). Ethical approval for work with primary DLBCL cells was obtained from the Ethical Commission of the Canton of Zurich (KEK-ZH-Nr. 2009-0062/1). Informed consent was obtained from all subjects, and all experiments using human material conformed to the principles set out in the WMA Declaration of Helsinki and the Department of Health and Human Services Belmont Report.

## Western blotting

Protein extracts were made in RIPA buffer (50 mM Tris–HCl, pH 8.0, 150 mM sodium chloride, 1% NP-40, 0.5% sodium deoxycholate, 0.1% SDS) supplemented with 2 mM sodium orthovanadate, 15 mM sodium pyrophosphate, 10 mM sodium fluoride, and 1× cOmplete protease inhibitor cocktail (Roche). Protein concentrations were determined using BCA assay (Pierce), and equal amounts were separated by SDS–PAGE (4–20% Mini-PROTEAN® TGX™ Precast Protein Gels; BIO-RAD) followed by transfer onto nitrocellulose membranes. Membranes were probed with antibodies against α-tubulin (DM1A, Sigma-Aldrich; 1:3,000 dilution), STAT3 (79D7, Cell Signaling Technology; 1:2,000), p-STAT3 (Tyr705, Cell Signaling Technology; 1:1,000), gp130 (E-8, Santa Cruz; 1:1,000), and β-actin (8H10D10, Cell Signaling Technology; 1:3,000).

## Flow cytometric analysis

Cells isolated from mouse spleens or bone marrow (jaw bones or femoral bones) were treated with ACK red blood cell lysis buffer pH 7.2–7.4 (150 mM $NH_4Cl$, 10 mM $KHCO_3$, 0.1 mM $Na_2EDTA$) and passed through a 40-μM cell strainer to produce single-cell suspensions. Cells were subsequently stained using the following fluorescent-labeled antibodies which were all purchased from Biolegend (clone numbers in brackets) and used at 1:100 dilution unless otherwise specified: Fixable Viability Dye eFluor 780 eBio/Affymetrix (1:1,000), and Hu FCR binding inhibitor purified, APC anti-human CD45 (2D1), PE anti-mouse CD45 (30-F11), APC anti-human CD19 (HIB19), PE-Cy7 anti-human NKp46 (9E2), and PB anti-human CD20 (HI47), all from Thermo; and purified rat anti-mouse CD16/CD32 (93), PB anti-mouse CD45 (30-F11), PB anti-human CD45 (HI30), PerCP-Cy5.5 anti-human CD33 (P67.6), BV605 anti-human CD3 (UCHT1), PE anti-STAT3 Phospho Tyr705 (13A3-1), PE anti-mouse IgG1κ (MOPC-21), APC anti-human Ig light chain lambda (MHL-38), PE anti-human Ig light chain lambda (MHL-38), PE anti-human Ig light chain kappa (MHK-49), FITC anti-human Ig light chain kappa (MHK-49), and PB anti-human CD19 (HIB19), all from Biolegend. For immunophenotyping of DLBCL cell lines, cells were stained with APC anti-human CD130 (2E1B02), APC mouse IgG2a kappa isotype ctrl (MOPC-173), APC anti-human CD126 (UV4), PE anti-human CD126 (UV4), APC mouse IgG1 kappa isotype ctrl (MOPC-21), PE mouse IgG1 kappa isotype ctrl (MOPC-21), and PE anti-STAT3 Phospho Tyr705 (13A3-1; 1:10 dilution) (all from Biolegend). Data were acquired on CyAn ADP (Beckman Coulter) or LSRFortessa (BD) flow cytometers and analyzed with the FlowJo software package (TreeStar).

## Quantitative RT–PCR

RNA was extracted using the NucleoSpin RNA kit (Macherey-Nagel). One microgram of total RNA was reverse-transcribed using SuperScript III reverse transcriptase (Invitrogen). For qRT–PCR, LightCycler 480 SYBR Green Master I (Roche) or TaqMan Gene Expression Assays (Thermo Fisher Scientific) were used, followed by analysis on a LightCycler 480 instrument. Samples were measured in duplicate. For the SYBR primer pairs, the efficiency was calculated by performing dilution series experiments. Target mRNA abundance was subsequently calculated relative to human RPLP32. The primers used were as follows: RPLP32, fwd: GAA GTT CCT GGT CCA CAA CG, rev: 5′-GCG ATC TCG GCA CAG TAA G; and IL6-R, fwd: CAC ATT CCT GGT TGC TGG AG, rev: GCT TCC ACG TCT TCT TGA ACC. For TaqMan assays, proprietary probes

**The paper explained**

**Problem**

Diffuse large B-cell lymphoma is an aggressive and often fatal B-cell malignancy for which targeted therapies other than the CD20-specific antibody rituximab are not available. Large-scale multi-omics studies have recently shed light on the mutational, epigenetic, and transcriptional landscapes of the disease. The development of animal models in which cell lines or patient samples could be screened pre- or co-clinically for drug susceptibilities, and in which the relevance of aberrantly inactivate or hyperactive signaling pathways could be examined, is woefully lagging behind.

**Results**

In this work, we describe a useful and highly reproducible model of DLBCL cell line and primary cell xenotransplantation that exploits genetically humanized "MISTRG" mice, which harbor human knock-in alleles encoding cytokines and growth factors for various human leukocyte subsets. We further humanized MISTRG mice by reconstituting them at birth with a normal human immune system, which required their transplantation with human CD34$^+$ hematopoietic stem and progenitor cells. MISTRG mice that had been humanized in this manner or that had additionally been genetically modified to express human IL-6 from a knock-in allele turned out to be even better hosts for DLBCL cell lines and primary cells, which, upon intravenous injection, engrafted in the bone marrow, spleen, and non-lymphoid tissues such as kidneys, liver, and lung. Labeling of our cell lines with luciferase further allowed us to track them in their murine hosts over time.

The contribution of IL-6 raised the question whether DLBCL cell lines generally require this cytokine for growth. In the course of extensive studies *in vitro* and *in vivo*, we found that most DLBCL cell lines of the (histologically specified) ABC-DLBCL subtype express both chains of the IL-6 receptor, with some of them even producing their own cytokine to stimulate an autocrine pro-proliferative loop. Others had inactivating mutations in a negative regulator of IL-6 signaling, SOCS1, which explained why they had become independent of exogenous addition of the cytokine. We were indeed able to confirm that a cell line producing IL-6 did not benefit from, or require, human IL-6 expression for growth in MISTRG mice. Similar findings were obtained in genetically engineered mice that lack IL-6 signaling specifically in the B-cell compartment and are largely protected from aberrant B-cell hyperproliferation and lymphomagenesis.

**Impact**

We propose that our findings have clinical impact for patients with DLBCL that is positive for the IL-6 receptor and also positive for phosphorylated STAT3, which acts downstream of IL-6 signaling and has been known for a number of years to be a strong negative predictor of DLBCL patient survival.

An IL-6R-specific antibody, named tocilizumab, exists and is approved and marketed for human use in patients with rheumatoid arthritis and in patients suffering from cytokine release syndrome after CAR T-cell therapy. We used this antibody to demonstrate in our lymphoma-bearing mice that blocking the IL-6R slows tumor growth and comes with virtually no side effects to the mice. We believe that DLBCL patients with IL-6R-positive, STAT3-positive DLBCL that are wild type for SOCS1 and perhaps also produce IL-6 should qualify to receive this antibody, ideally as a combination therapy with other targeted agents that are emerging from clinical trials. In summary, we propose that targeting the IL-6 signaling pathway represents an opportunity for personalized treatment that should be explored in clinical trials in DLBCL patients.

spanning exons 13–14 of the IL-6 signal transducer/gp130 (IL6ST: Hs00174360_m1) and binding to exon 1 of actin (ACTB: Hs99999903_m1), respectively, were used.

## Immunohistochemistry

Cumulative data of three independent DLBCL study cohorts were used for *in silico* analysis of the prognostic role of p-STAT3 expression. All cases have been stained utilizing the Cell Signaling antibody clone D3A7 mAb9145 at a 1:50 dilution. The cutoff to assign positivity has been set at > 17% staining tumor cells as estimated by receiver operating characteristic analysis taking survival as a set variable. The antibody used for phospho-JAK2 (Tyr1007/1008) staining was the Cell Signaling antibody clone C80C3 mAb3776 (1:50 dilution), and the cutoff to assign positivity has been set at > 2.5% positively staining tumor cells. Overall and disease-specific survival were measured from initial diagnosis to death of any cause or on/with lymphoma, respectively, or last follow-up. The probability of survival was determined using the Kaplan–Meier method, and differences were compared using the log-rank with the help of the Statistical Package for the Social Sciences (IBM SPSS version 22.0, Chicago, IL, USA). Two of the above study collectives were stained for the IL-6R subunit gp130 applying the Abcam polyclonal antibody ab170257 at a 1:200 dilution. The cutoff to assign positivity was set at > 30% positively staining tumor cells taking p-STAT3 positivity as a set variable. The correlation between expression of both p-STAT3 and gp130 was estimated using the chi-square test.

## Statistics

All statistical analyses were performed using GraphPad Prism software. Graphs represent means plus SD of at least two independent experiments, and statistical analysis was performed using two-tailed Mann–Whitney test for *in vivo* studies and correlation analysis was performed using one-way ANOVA. A log-rank test was performed for comparison of survival curves.

**Expanded View** for this article is available online.

## Acknowledgements

The authors wish to thank all consortium members of the Clinical Research Priority Program for support and helpful discussions. This study was funded by the Swiss Cancer Leagues grants KLS-3612-02-2015 and KFS-4120-02-2017 to A.M. Additional support was provided by the Clinical Research Priority Program "Human Hemato-lymphatic Diseases" of the University of Zurich. H.H. was supported by the Forschungskredit of the University of Zurich. A.P.A.T. is supported by the Prof. Dr. Max Cloëtta Foundation.

## Author contributions

HH and KB designed, performed, and analyzed experiments and co-wrote the article. AS, KS, SK, and C-TW helped with experiments and provided critical tools and advice. APAT and MGM provided critical tools and patient samples. AT stained, analyzed, and provided patient samples, and AM supervised the study and co-wrote the article.

## Conflict of interest

The authors declare that they have no conflict of interest.

## For more information

https://www.lymphoma.org/aboutlymphoma/nhl/dlbcl/
https://www.leukaemia.org.au/disease-information/lymphomas/

non-hodgkin-lymphoma/other-non-hodgkin-lymphomas/diffuse-large-b-cell-lymphoma/
https://www.mayoclinic.org/diseases-conditions/lymphoma/symptoms-causes/syc-20352638

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
