## [Review Process File · EMBO Molecular Medicine]

The IL-6 signaling complex is a critical driver, negative prognostic factor and therapeutic target in diffuse large B-cell lymphoma

Hind Hashwah, Katrin Bertram, Kristin Stirm, Anna Stelling, Cheuk-Ting Wu, Sabrina Kasser, Markus G. Manz, Alexandre P.A. Theocharides, Alexandar Tzankov and Anne Müller

Review timeline:

Submission date:	8 March 2019
Editorial Decision:	4 April 2019
Revision received:	5 July 2019
Editorial Decision:	8 August 2019
Revision received:	22 August 2019
Accepted:	23 August 2019

Editor: Lise Roth

Transaction Report:

1st Editorial Decision

4 April 2019

Thank you for the submission of your manuscript to EMBO Molecular Medicine. We have now heard back from the referees whom we asked to evaluate your manuscript.

As you will see from the reports below, the 3 referees acknowledge the potential interest of the findings for the field, however they also have fundamental concerns that should be addressed in a major round of revision of the present manuscript. In particular, the mechanism should be strengthened in order to fully support the conclusions (thorough characterization of IL-6 signalling pathway, IL-6R expression analyses).

Addressing the reviewers' concerns in full will be necessary for further considering the manuscript in our journal. EMBO Molecular Medicine encourages a single round of revision only and therefore, acceptance or rejection of the manuscript will depend on the completeness of your responses included in the next, final version of the manuscript.

Please also contact us as soon as possible if similar work is published elsewhere. If other work is published, we may not be able to extend the revision period beyond three months.

I look forward to receiving your revised manuscript.

***** **Reviewer's comments** *****

Referee #1 (Remarks for Author):

In their current manuscript, Hashwah et al. describe the validation and improvement of their MISTRG mouse model for the analysis of lymphomagenesis. They show that DLBCL cell lines and DLBCL cells derived from patients that express IL-6 receptor subunits engraft and result in orthotopic lymphomas in humanized IL-6 expressing mice (MISTRG6) and in mice with a reconstituted human immune system. Moreover, they demonstrate that expression of both IL-6 receptor subunits, IL-6 itself or mutational inactivation of SOCS1, are mechanisms through which a subset of DLBCLs achieve constitutive activation of the IL-6 pathway. Based on their overall data, the authors conclude that the ABC-DLBCL subtype preferentially but not exclusively displays activation of IL-6R/STAT3 signaling. In addition, they validate pSTAT3 positivity as a strong negative prognostic factor associated with lower overall- and disease specific survival of DLBCL patients independent of subtype classification. Using a mouse model for Myc-driven lymphomagenesis, they show that knockout of the *Il6ra* alleles specifically in the GC B-cell compartment delays tumor growth and reduces tumor burden. Finally, to address the therapeutic aspect of their findings, the authors treated MISTRG or MISTRG6 mice suffering from IL-6 signaling positive lymphomas with the neutralizing IL-6R antibody tocilizumab, which reduced lymphoma growth.

In general, this study is a nice and overall clearly presented piece of work using sophisticated mouse models that appear to be useful for the lymphoma community. Their finding that there is a DLBCL subgroup displaying active IL-6 signaling and that interference with this signaling pathway impairs tumor growth is of interest; however, there are various published studies that address the interleukin/JAK/STAT pathway in the lymphoma context (see for example PMID: 18160665, 25124685, 1472922, 16703758, 16892169, 28860565, 22968454). Another major shortcoming of this study is the poor characterization of the IL-6 signaling pathway in lymphoma cells. Points that need to be addressed prior to publication are:

Major points:

1. The authors have evidence that IL6 signaling plays an important role in their lymphoma models; to further validate this, the authors should assess the activity of this pathway (for example by western blotting of pJAK, pSTAT etc). In this respect figure 6D needs a positive control for pSTAT3.
2. The authors show that IL6 stimulates lymphoma growth in vivo. Is this effect diminished by knocking-down the IL6R in these cells?
3. The authors state that they could not detect gp130 protein expression in the DLBCL cells via flow cytometry (page 11, line 6). Did they try to detect it via Western Blotting?
4. Unfortunately the authors could stain the formalin-fixed patient material for gp130 only but not for the IL-6R α -chain. Maybe it would be an option to stain the patient material for IL-6 itself or for phospho-marks in this pathway to gain more insights into the importance of the IL-6 signaling axis in DLBCL patients.
5. It is a bit disappointing that the observed differences upon tocilizumab treatment are not statistically significant in 2 of 3 in vivo scenarios (figure 7 G/I/J). Do the authors have any explanation for this? Given the interesting trend they should consider to treat more mice to make a more definitive statement. Did the authors try any drug combinations to increase tocilizumab efficacy?
6. It is hard to understand how the numbers for tissue involvement in figures 1D, 2I and 3D were determined on the basis of whole-body analysis? The authors need to add a paragraph to the M&M section in order to address this.
7. In the M&Ms, the RC-K8 cell line is listed to be of the GCB subtype. However, in Figure 5 B/C and the respective figure legend, the authors state that the RC-K8 cell line is unclassified and in the text, it is written that 'this cell line has features of both ABC and GCB subtypes' (page 11, line 8/9). For the latter, it is not clear whether this is just their conclusion from the observation that the RC-K8 cells express CD126 and gp130 or whether this is the reason for expression of both subunits. This issue might be confusing for the reader and needs clarification.
8. Please include in the M&M section from which sources the cell lines were obtained and whether they were authenticated. For the interested reader, it would be nice to have a reference given for the cell of origin classification of the different cell lines.

Minor points:

1. Full name for the SU-DHL cell lines needs to be given in Figure 1A. For most of them 'SU' is missing.
2. In Figure 2C, there is no definition for the 2 differently colored graphs given, neither in the figure itself nor in the figure legend.
3. The overall size of immunohistochemistry image in Figure 4H is too small and need to be increased.
4. The size of the scale bar for figures 4H, 5D, 6F and 7H need to be given in the figure legend since the labeling is not readable in the figure itself.
5. Please indicate in the figure legend for Figure 5B/C what 'Buffy B' stands for (most probably B cells that were isolated from buffy coats from healthy specimen).
6. Please indicate Manufacturer for Nucleospin RNA kit in the supplemental M&M section 'Quantitative RT-PCR'
7. For the tumor volume over time in Supplemental Fig. 1, please include the number of animals in the figure legend.
8. In some subfigures, the text is not readable due to small font size or poor quality of the image (e.g. Figure 3A/B, Figure 7B). Please assure that text is readable or remove text if it is not necessary figure content. Please assure good quality of all figures

Referee #2 (Remarks for Author):

The manuscript entitled "The IL-6 signaling complex is a critical driver, negative prognostic factor and therapeutic target in diffuse large B-cell lymphoma" describes how IL6-signaling is required for the growth of a subset of non-GBC DLBCL cells. Using humanized MISTRG mouse lines, the authors first described the tissue distribution of established DLBCL cell lines and later showed how a MISTRG-based mouse strain that expresses human IL6 (hIL6), MISTRG6 was used to enhance in vivo DLBCL growth. The authors subsequently showed how MISTRG6 but not MISTRG supported the growth of primary human DLBCL cells and that IL6 receptor (IL6Ra), SOCS1, and pSTAT3 are all relevant to the hypothesis that IL6-signaling pathway regulates the in vivo survival of DLBCL cell lines. Finally, the authors showed how IL6 signaling is supportive of the MYC-driven DLBCL mouse model and how the IL6-dependent subset of DLBCL can be targeted for antibody therapy. Overall, the idea of a non-GBC DLBCL cell population that is hIL6-dependent and hence can be targeted for IL6-antibody therapy is substantial but the manuscript does require more supportive data. The manuscript needs several clarifications to support their hypothesis and below are a few suggestions:

1. While the gender of the animals used was stated in Figure 1a-c, it is unclear in the subsequent experiments. Since there is a clear engraftment difference on male vs female, the authors should clarify this and keep the difference in mind. If only one gender is used in the following experiments, it should be clearly stated. Or, if both genders were used, the lack of difference should be explained.
2. For figure 2, it is unclear as to what HSC reconstitution contributed to support DLBCL engraftment. Following the rationale of the manuscript, maybe the systemic level of IL6 should be measured and reported. Alternatively, it will be interesting to see if infusion of IL6 into the mice would recapitulate the effect of HSC reconstitution.
3. In figure 4, primary human DLBCL cells were first expanded in MISTRG6 mice and this would have potentially selected cells that are IL6-dependent. It is then no surprise that they failed to expand in MISTRG and therefore undermined the significance of the study. Perhaps the author should supplement IL6 to the MISTRG mice to see if primary DLBCL cells will expand like the MISTRG6.

Referee #3 (Remarks for Author):

In the following manuscript, Hashwah and Bertram et. al. examine the role of IL-6 signaling in DLBCL pathogenesis, evaluate its prognostic power, and propose IL-6 targeting as a potential therapeutic strategy for lymphomagenesis. The authors first assessed the ability of DLBCL cell lines to engraft into genetically humanized animal strain MISTRG, that expresses human cytokines (M-CSF, IL-3, GM-CSF, THPO) and hSIRP α in the Rag $^{-/-}$ IL-2R $\gamma^{-/-}$ background and find that although lymphoma cell lines reconstituted the animals, engraftment efficiency was significantly better if MISTRG animals haven been previously reconstituted with hematopoietic

stem cells or expressed human IL-6 (MISTRG6). Importantly, the authors demonstrate that MISTRG animals reconstituted with human hematopoietic cells and MISTRG6 hosts support engraftment of primary DLBCL samples, a significant finding as it expands scientific toolkit to study primary lymphoma cells. Next, the authors touched up on the mechanisms of STAT3 activation on DLBCL animals and prognostic value of pSTAT3. Finally, the group assessed therapeutic potential of tocilizumab, a neutralizing antibody targeting IL6R and showed that animals treated with tocilizumab showed a significant reduction in lymphoma burden.

I find the studies using MISTRG6 animals and DLBCL xenografts informative and believe authors' findings are highly beneficial to the field overall. My main concern is apparent lack of IL6R surface expression on majority of examined DLBCL cell lines, suggesting that the effect of IL-6 on lymphoma cells, the authors describe, might not be a direct. My second concern is the authors' claims regarding STAT3 activation in different DLBCL samples. A considerably more in-depth analysis is needed to support conclusions the authors make. Overall, I believe the manuscript will significantly benefit if the authors expand on the findings detailing expansion of primary DLBCL cells in MISTRG6 animals and demonstrate surface levels of IL6R on multiple primary DLBCL samples.

General Points:

1. Fig 4G. Please include analysis of IL6R surface levels on at least three primary DLBCL samples.
2. Please clearly indicate how many different primary xenografts were tested for engraftment in MISTRG6 animals.
3. Figure 7 E. Please perform statistical analysis.
4. Please increase the font size in the figures. It is extremely difficult to read the labels.

Specific points:

1. The hIL6R expression profiles of RC-K8 and U-2932 cells appears to be duplicated between Figure 3F and Figure 5A. Please remove the duplication and clearly indicate how many times the staining was done.
2. Please add scale bars to figures 4C and make sure that the scale bars (and the numerical designation associated with them) are visible.

1st Revision - authors' response

5 July 2019

***** Reviewer's comments *****

General remarks

We would like to thank our reviewers for thoughtful and useful comments, which we have now addressed experimentally throughout. In particular, we have generated an IL-6R ko cell line and tested it side by side with the parental cell line for engraftment, have repeated the tocilizumab treatment in several more experiments, and have focused more attention on the genetically modified IL-6R ko mouse model. All points raised by our reviewers have been addressed with more experimentation. Please see the point-by-point response to our reviewers' comments below. We also added the following text and data on the genetically modified IL-6R ko mouse model although it was not specifically requested by any reviewer; however we believe the additional data strengthens our claims:

Results, p. 14: However, the majority of mice harboring *MYC* developed lymphomas in their axillary and inguinal lymph nodes and spleen, which was somewhat (albeit not significantly) delayed by the genetic ablation of both copies of the *Il6ra* (Fig 7A). Heterozygous mice with one functional copy of *Il6ra* behaved like wild type mice in this setting (Figure 7A). IL-6ra deficiency in B-cells resulted not only in a delay, but also in a reduction of the tumor burden, which manifested in significantly smaller lymph nodes at the study endpoint (Fig 7B). To address whether the hyperproliferation of B-cells undergoing the GC reaction upon sheep red blood cell immunization was affected by loss of IL-6ra, we immunized *AID-Cre x Il6ra^{f/f}* mice (not harboring the *MYC* transgene) and their heterozygous and wild type littermates and quantified their GC compartment by flow cytometry and Ki67 immunohistochemistry. The immunization led to a strong increase in CD19⁺CD95⁺CD38^{low} GC B-cells as determined by flow cytometry, which was reduced by the

deletion of one, and especially of both alleles of *Il6ra* (Fig 7C). The smaller GC compartment of *AID-Cre x Il6ra^{fl/fl}* mice could be attributed to reduced frequencies of CXCR4^{hi}CD86^{low} centroblasts, whereas CXCR4^{low}CD86^{low} centrocytes were unchanged (Fig 7D and E). We further stained spleen sections for Ki67 (Fig 7F) and quantified the GC numbers and area; this approach revealed that the multiplicity of individual GCs as well as their total area was reduced in mice lacking one or both alleles of *Il6ra* (Fig 7G-I). The combined results indicate that IL-6 signaling is GC B-cell-intrinsically required for early onset, aggressive lymphomagenesis.

Also, we have replaced the OCI-Ly10 data in Figure 1 with a similar dataset for the RIVA cell line. The gender bias (higher tumor burden in females than in males) of OCI-Ly10 cells is not representative of DLBCL cell lines; the results obtained with OCI-Ly10 therefore were a bit misleading. The RIVA data are more appropriate to show here.

Referee #1 (Remarks for Author):

In their current manuscript, Hashwah et al. describe the validation and improvement of their MISTRG mouse model for the analysis of lymphomagenesis. They show that DLBCL cell lines and DLBCL cells derived from patients that express IL-6 receptor subunits engraft and result in orthotopic lymphomas in humanized IL-6 expressing mice (MISTRG6) and in mice with a reconstituted human immune system. Moreover, they demonstrate that expression of both IL-6 receptor subunits, IL-6 itself or mutational inactivation of SOCS1, are mechanisms through which a subset of DLBCLs achieve constitutive activation of the IL-6 pathway. Based on their overall data, the authors conclude that the ABC-DLBCL subtype preferentially but not exclusively displays activation of IL-6R/STAT3 signaling. In addition, they validate pSTAT3 positivity as a strong negative prognostic factor associated with lower overall- and disease specific survival of DLBCL patients independent of subtype classification. Using a mouse model for Myc-driven lymphomagenesis, they show that knockout of the *Il6ra* alleles specifically in the GC B-cell compartment delays tumor growth and reduces tumor burden. Finally, to address the therapeutic aspect of their findings, the authors treated MISTRG or MISTRG6 mice suffering from IL-6 signaling positive lymphomas with the neutralizing IL-6R antibody tocilizumab which reduced lymphoma growth.

In general, this study is a nice and overall clearly presented piece of work using sophisticated mouse models that appear to be useful for the lymphoma community. Their finding that there is a DLBCL subgroup displaying active IL-6 signaling and that interference with this signaling pathway impairs tumor growth is of interest; however, there are various published studies that address the interleukin/JAK/STAT pathway in the lymphoma context (see for example PMID: 18160665, 25124685, 1472922, 16703758, 16892169, 28860565, 22968454). Another major shortcoming of this study is the poor characterization of the IL-6 signaling pathway in lymphoma cells. Points that need to be addressed prior to publication are:

Major points:

1. The authors have evidence that IL6 signaling plays an important role in their lymphoma models; to further validate this, the authors should assess the activity of this pathway (for example by western blotting of pJAK, pSTAT etc). In this respect figure 6D needs a positive control for pSTAT3.

The experiment shown in Figure 6D has been repeated and the Western blot has been rerun with a positive control included. The WB in panel 6D has now been exchanged for the new one. To address which JAK is important in DLBCL downstream of the IL-6R, we purchased a total of 5 different JAK1 and JAK2, as well as pJAK1 and pJAK2 antibodies and probed our extracts shown in Figure 6B and 6D with all of these. While it is clear that both JAK1 and JAK2 are expressed, none of the phospho-specific JAK antibodies gave us trustworthy results. We therefore resorted to the IHC staining for pJAK2, which is reliable and has been shown in the past to correlate with active STAT3 signaling in DLBCL. We reanalyzed our patient samples for a pJAK2/pSTAT3 correlation and are showing this data in the form of the new supplemental Figure EV5. The text has been modified as follows:

Results, p.15: pSTAT3 positivity was a strong negative prognostic factor in the entire cohort (Fig 8B) and associated with higher overall- as well as disease-specific mortality in both ABC- and GCB-DLBCL subtypes (Fig 8C and D). To address whether active STAT3 signalling correlated with activity of the JAK2 kinase, which has previously been implicated in STAT3 phosphorylation in DLBCL (25), we selected 330 cases of which roughly one half was pSTAT3-positive and the other was pSTAT3-negative. All cases were evaluable by immunohistochemistry for activated (phosphorylated at Tyr1007/1008) JAK2 (Fig EV5A). Regression analysis revealed a positive association between pSTAT3 and pJAK2 (Fig EV5B). Of pSTAT3-positive cases, the vast majority was also positive for pJAK2; however, we also detected a large number of pJAK2-positive cases that were negative for pSTAT3, indicating that additional factors (STAT3 expression or subcellular localization for example) may affect STAT3 phosphorylation (Fig EV5C). The combined results indicate that the IL-6R/STAT3 signalling axis is preferentially but not exclusively active in ABC-DLBCL, and a negative prognosticator for both subtypes; STAT3 phosphorylation correlates with activation of the upstream kinase JAK2.

2. The authors show that IL6 stimulates lymphoma growth in vivo. Is this effect diminished by knocking-down the IL6R in these cells?

This is a very interesting suggestion. We have now used CRISPR to generate RC-K8 cells that lack IL-6R and therefore cannot respond to IL-6. Indeed, we find that STAT3 phosphorylation is abrogated in IL-6R ko cells upon IL-6 addition. Furthermore, IL-6R ko RC-K8 cells colonize the bone marrow of MISTRG6 mice less efficiently than their wild type counterparts. These data are now all shown in supplemental figures and the text has been revised as follows:

Results, p.9: To address whether the growth advantage of RC-K8 cells in MISTRG6 mice could indeed be attributed to CD126 expression, we targeted the second exon of the corresponding gene *IL6R* by CRISPR, which resulted in the complete loss of CD126 surface expression (FigEV3B). RC-K8 cells that had lost CD126 expression colonized MISTRG6 bone marrow less efficiently than their wild type counterparts (Fig EV3B). The differential benefit of the two examined DLBCL cell lines from IL-6 provision mirrors the benefit conferred by human immune cell reconstitution;...

and

Results, p.12ff: Interestingly, the two (ABC subtype) cell lines SU-DHL-2 and OCI-Ly3 that produce IL-6 and express gp130 exhibit constitutive phosphorylation and activation of STAT3 as assessed by Western blotting and flow cytometric analysis of STAT3 phosphorylation on tyrosine 705 (Figure 6B, C). In contrast, all other ABC-DLBCL cell lines (with one exception) that express gp130 activate STAT3 only upon exposure to exogenous IL-6 (Figure 6B). None of the GCB-DLBCL cell lines show constitutive STAT3 activation or respond to IL-6 *in vitro* despite producing STAT3 at similar levels as the ABC-DLBCL cell lines (Figure 6D). RC-K8 cells lacking the IL-6R a-chain CD126 due to a genomic editing by CRISPR (Fig EV3A) fail to phosphorylate STAT3 upon addition of IL-6 (Fig 6E).

3. The authors state that they could not detect gp130 protein expression in the DLBCL cells via flow cytometry (page 11, line 6). Did they try to detect it via Western Blotting?

After unsuccessfully testing various gp130 antibodies for WB application, we finally managed to obtain trustworthy signals with one of them. A representative WB, along with its quantification, is now shown in the new supplemental Figure 4. The text has been modified as follows:

Results, p.11ff: The expression of gp130 was difficult to detect by flow cytometry; qRT-PCR results indicate that gp130 is weakly expressed by all ABC-DLBCL cell lines, but not by GCB-DLBCL cell lines (Figure 5C). This observation could be confirmed by Western blotting (Fig EV4A).

4. Unfortunately the authors could stain the formalin-fixed patient material for gp130 only but not for the IL-6R a-chain. Maybe it would be an option to stain the patient material for IL-6 itself or for phospho-marks in this pathway to gain more insights into the importance if the IL-6 signaling axis in DLBCL patients.

We are including quantitative pSTAT3 data in various sections of the manuscript. This is now backed up by pJAK2 staining, which was performed on 330 cases. The data is included in suppl.

Figure 7 and mentioned in the text as follows. Staining for IL-6 itself was unfortunately not possible, although this information would have been nice to have for a large cohort of patients.

Results, p.15: pSTAT3 positivity was a strong negative prognostic factor in the entire cohort (Fig. 8B) and associated with lower overall- as well as disease-specific survival in both ABC- and GCB-DLBCL subtypes (Figure 8C, D). To address whether active STAT3 signaling correlated with activity of the JAK2 kinase, which has previously been implicated in STAT3 phosphorylation in DLBCL (24), we selected 330 cases of which roughly one half was pSTAT3-positive and the other was pSTAT3-negative. All cases were evaluable by immunohistochemistry for activated (phosphorylated at Tyr1007/1008) JAK2 (Fig EV5A). Regression analysis revealed a positive association between pSTAT3 and pJAK2 (Fig EV5B). Of pSTAT3-positive cases, the vast majority was also positive for pJAK2; however, we also detected a large number of pJAK2-positive cases that were negative for pSTAT3, indicating that additional factors (STAT3 expression or subcellular localization for example) may affect STAT3 phosphorylation (Fig EV5C). The combined results indicate that the IL-6R/STAT3 signaling axis is preferentially but not exclusively active in ABC-DLBCL, and a negative prognosticator for both subtypes; **STAT3 phosphorylation correlates with activation of the upstream kinase JAK2.**

5. It is a bit disappointing that the observed differences upon tocilizumab treatment are not statistically significant in 2 of 3 in vivo scenarios (figure 7 G/I/J). Do the authors have any explanation for this? Given the interesting trend they should consider treating more mice to make a more definitive statement. Did the authors try any drug combinations to increase tocilizumab efficacy?

This reviewer is right in pointing out that the tocilizumab data are not statistically significant in some settings. We indeed see the best effects in a cell line that produces its own IL-6, OCI-Ly3; this cell line appears to depend most on IL-6. We have now included more datapoints on tocilizumab-treated MISTRG6 mice harboring primary DLBCL cells or RC-K8 cells. Pooled data from 2 and up to 3 studies are now shown in the panels in question. Although the results for primary cells and the RC-K8 cell line are still not significant ($p > 0.05$), the trend is clearer now in all three settings. We have not performed any drug combinations, although this would be interesting to do in the future. As the responsive cell lines are mostly of the ABC-DLBCL subtype, we would expect to see synergistic effects with ibrutinib or lenalidomide. This is definitely on the to-do list for future work. The text has been modified as follows:

Results, p.16: Mice received either an isotype control antibody or tocilizumab starting two weeks after lymphoma cell transplantation. Mice on tocilizumab showed a reduction of the spleen size at the study endpoint (Fig 8E); the phosphorylation of STAT3, assessed immunohistochemically on paraffin-embedded material, was reduced in response to the treatment (Fig 8F). We next examined the effects of tocilizumab on MISTRG mice transplanted with the cell line OCI-Ly3, which produces IL-6 (Fig 6A). Indeed, (non-IL-6-expressing) MISTRG mice harboring OCI-Ly3 cells exhibited detectable levels of hIL-6 in their serum (Fig EV5D). Tocilizumab treatment reduced the OCI-Ly3 tumor burden in the bone marrow (Fig 8G). In a third experimental setting, MISTRG6 mice transplanted with RC-K8 cells were subjected to tocilizumab treatment, which also detectably reduced the tumor burden (Fig 8H). The combined results from our patient cohort and xenotransplantation models indicate that active IL-6R/STAT3 signaling is a negative prognostic factor and therapeutic target in DLBCL; biomarkers that may be useful in predicting therapy success include the expression of IL-6R, gp130 and phospho-STAT3.

6. It is hard to understand how the numbers for tissue involvement in figures 1D, 2I and 3D were determined on the basis of whole-body analysis? The authors need to add a paragraph to the M&M section in order to address this.

We are now showing representative IVIS images of organs that were visualized for luciferase expression ex vivo upon euthanasia of the mice in suppl. Figure 1. We have added the following statements to the Results and Materials and Methods section to explain the procedure.

Results, p.6: A careful whole-body analysis through a combination of IVIS of explanted organs (representative IVIS images shown in Fig EV1) with macroscopic inspection and histology at the study endpoint further revealed that all three cell lines rather consistently colonize the lungs, kidneys, liver and reproductive tract of both male and female MISTRG mice (Figure 1D).

Materials and Methods, p.22: For whole-body bioluminescent imaging of mice included in the orthotopic model, mice were i.p. injected with D-luciferin (150mg kg^{-1}) (VivoGlo Luciferin, Promega) and mice were analyzed 10 minutes later using an IVIS Spectrum system (Caliper LifeSciences). Bioluminescence images were acquired using 'auto' setting with F/stop = 1 and binning = medium. A digital false-color photon emission image of the mouse was generated, and photons were counted within the whole-body area. Photon emission was measured as radiance in $\text{p.s}^{-1}.\text{cm}^{-2}.\text{sr}^{-1}$. At the study end point of orthotopic models, spleens were harvested and weighed, and hind legs/jaw bones were collected for the analysis of the bone marrow. **Tissue colonization by lymphoma cells was assessed for the liver, kidney, lung and reproductive organs after removal from the carcass using two complementary approaches: livers and kidneys were first macroscopically inspected for the appearance of white nodules indicative of tumor growth, and then placed individually, alongside the lung lobes and ovaries or testicles into 12-well plates filled with luciferin in PBS ($300\mu\text{g/ml}$) for *ex vivo* IVIS imaging. Organs were imaged within 5 min of euthanasia.**

7. In the M&Ms, the RC-K8 cell line is listed to be of the GCB subtype. However, in Figure 5 B/C and the respective figure legend, the authors state that the RC-K8 cell line is unclassified and in the text, it is written that 'this cell line has features of both ABC and GCB subtypes' (page 11, line 8/9). For the latter, it is not clear whether this is just their conclusion from the observation that the RC-K8 cells express CD126 and gp130 or whether this is the reason for expression of both subunits. This issue might be confusing for the reader and needs clarification.

RC-K8 are debated in the literature. The German cell line repository DSMZ sells them as GCB-subtype. However, Kalaitzidis and colleagues demonstrated in 2002 that the RC-K8 cell line displays dysregulated NF- κ B signaling which is a defining feature of the ABC subtype. This paper is now cited and the issue discussed. In our experimental system, RC-K8 show clear similarities to ABC subtype cell lines.

8. Please include in the M&M section from which sources the cell lines were obtained and whether they were authenticated. For the interested reader, it would be nice to have a reference given for the cell of origin classification of the different cell lines.

All cell lines were recently authenticated and described in detail (also with respect to their cell or origin classification) by our collaborator Alexandar Tzankov in Juskevicius et al, 2017, along with their mutational profiles and sources. This is now mentioned in the M&M section.

Materials and Methods, p.21: Cell culture experimentation and ELISA. The DLBCL cell lines used here included six of GCB-DLBCL subtype (SU-DHL-4, SU-DHL-5, SU-DHL-6, SU-DHL-8, SU-DHL-10, SU-DHL-16) and five of ABC DLBCL subtype (U-2932, OCI-Ly3, OCI-Ly10, SU-DHL2, and RIVA) and one unclassified cell line (RC-K8) that have all been described previously (15, 16). **Cell line authentication was performed according to the guidelines of the International Cell Line Authentication Committee using short tandem repeat profiling; the result of this effort was described recently for all 12 cell lines used here, along with their sources (22).**

Minor points:

1. Full name for the SU-DHL cell lines needs to be given in Figure 1A. For most of them 'SU' is missing.

There is unfortunately no space to spell out SU-DHL for all cell lines, especially since we had to substantially increase the font size. Therefore, we have used the full name once, and the abbreviation -DHL- in the rest of the figure panel.

2. In Figure 2C, there is no definition for the 2 differently colored graphs given, neither in the figure itself nor in the figure legend.

This is an oversight. We are now explaining the color code in the figure itself.

3. The overall size of immunohistochemistry image in Figure 4H is too small and need to be increased.

Has now been done for all IHC figure panels throughout the paper.

4. The size of the scale bar for figures 4H, 5D, 6F and 7H need to be given in the figure legend since the labeling is not readable in the figure itself.

We've now made sure to mention the size of each scale bar in the accompanying legend.

5. Please indicate in the figure legend for Figure 5B/C what 'Buffy B' stands for (most probably B cells that were isolated from buffy coats from healthy specimen).

This has been addressed in the appropriate legend (now a supplemental figure).

6. Please indicate Manufacturer for Nucleospin RNA kit in the supplemental M&M section 'Quantitative RT-PCR'

This has been addressed.

7. For the tumor volume over time in Supplemental Fig. 1, please include the number of animals in the figure legend.

The number of tumors per treatment group is now specified directly in the figure.

8. In some subfigures, the text is not readable due to small font size or poor quality of the image (e.g. Figure 3A/B, Figure 7B). Please assure that text is readable or remove text if it is not necessary figure content. Please assure good quality of all figures

This has now been taken care of. All figures have been redone with larger font size and higher resolution images.

Referee #2 (Remarks for Author):

The manuscript entitled "The IL-6 signaling complex is a critical driver, negative prognostic factor and therapeutic target in diffuse large B-cell lymphoma" describes how IL6-signaling is required for the growth of a subset of non-GBC DLBCL cells. Using humanized MISTRG mouse lines, the authors first described the tissue distribution of established DLBCL cell lines and later showed how a MISTRG-based mouse strain that expresses human IL6 (hIL6), MISTRG6 was used to enhance in vivo DLBCL growth. The authors subsequently showed how MISTRG6 but not MISTRG supported the growth of primary human DLBCL cells and that IL6 receptor (IL6Ra), SOCS1, and pSTAT3 are all relevant to the hypothesis that IL6-signaling pathway regulates the in vivo survival of DLBCL cell lines. Finally, the authors showed how IL6 signaling is supportive of the MYC-driven DLBCL mouse model and how the IL6-dependent subset of DLBCL can be targeted for antibody therapy. Overall, the idea of a non-GBC DLBCL cell population that is hIL6-dependent and hence can be targeted for IL6-antibody therapy is substantial but the manuscript does require more supportive data. The manuscript needs several clarifications to support their hypothesis and below are a few suggestions:

1. While the gender of the animals used was stated in Figure 1a-c, it is unclear in the subsequent experiments. Since there is a clear engraftment difference on male vs female, the authors should clarify this and keep the difference in mind. If only one gender is used in the following experiments, it should be clearly stated. Or, if both genders were used, the lack of difference should be explained.

We generally included mixed gender groups in experiments, and made sure to randomize both genders to the various treatment arms (in the case of tocilizumab treatment for example) to avoid gender biases in treatment responses. This is now mentioned in the Materials and Methods section. The OCI-Ly10 cell line shown in Figure 1A-C was a bit unusual as it indeed engrafted better in (lighter) females than in (heavier) males, which was not observed with our other DLBCL cell lines. We therefore decided to exchange the OCI-Ly10 data in all of Figure 1 for data obtained with the RIVA cell line, which showed no gender bias.

Materials and Methods, p.22: For xenotransplantation studies, cell lines were injected subcutaneously (10×10^6 cells in 150 μ l PBS) into both flanks of 6-8-week-old mixed gender MISTRG mice, or intravenously (10×10^6 cells in 100 μ l PBS) into mixed gender MISTRG or MISTRG6 mice in the orthotopic model.

2. For figure 2, it is unclear as to what HSC reconstitution contributed to support DLBCL engraftment. Following the rationale of the manuscript, maybe the systemic level of IL6 should be measured and reported. Alternatively, it will be interesting to see if infusion of IL6 into the mice would recapitulate the effect of HSC reconstitution.

This is an important suggestion. We have now used a human IL-6 ELISA to compare hIL-6 levels in serum from HSC-reconstituted MISTRG mice to hIL-6 levels in unreconstituted MISTRG and in MISTRG6 mice. The levels of hIL-6 upon reconstitution are generally somewhat lower than those of MISTRG6 mice, but hIL-6 is clearly detectable in all reconstituted animals. This data is now shown in the supplement to Figure 3, and the text has been modified accordingly. We also deleted the IL-6R α -chain as suggested by reviewer 1, and this data is now also included in new supplemental figure.

Results, p.9: To assess whether differential expression of the IL-6 receptor could possibly explain the differences in terms of IL-6 dependence observed between the two cell lines, we assessed its surface expression by flow cytometry. Indeed, only RC-K8, but not U-2932 cells strongly expressed the IL-6 receptor α -chain CD126 (Fig 3F). To address whether the growth advantage of RC-K8 cells in MISTRG6 mice could indeed be attributed to CD126 expression, we targeted the second exon of the corresponding gene *IL6R* by CRISPR, which resulted in the complete loss of CD126 surface expression (Fig EV3A). RC-K8 cells that had lost CD126 expression colonized MISTRG6 bone marrow somewhat less efficiently than their wild type counterparts (Fig EV3B). The differential benefit of the two examined DLBCL cell lines from IL-6 provision thus mirrors the benefit conferred by human immune cell reconstitution (Fig 2). Indeed, we are able to detect hIL-6 in the serum of HSPC-reconstituted mice, albeit at lower levels than in serum from MISTRG6 mice (Fig EV3C). Surface expression of the IL-6 receptor CD126 appears to contribute to the differential growth of RC-K8 and U-2932 cells in MISTRG and MISTRG6 mice.

3. In figure 4, primary human DLBCL cells were first expanded in MISTRG6 mice and this would have potentially selected cells that are IL6-dependent. It is then no surprise that they failed to expand in MISTRG and therefore undermined the significance of the study. Perhaps the author should supplement IL6 to the MISTRG mice to see if primary DLBCL cells will expand like the MISTRG6.

The supplementation of IL-6 is in principle doable, but would have cost a fortune! The treatment would have had to continue for many weeks; therefore, we decided to address this point of our reviewer in other ways. We now show data on the hIL-6 levels in reconstituted MISTRG mice and MISTRG6 mice (similar levels; see above), and have attempted to transplant and engraft additional patient samples (of which we determined the IL6R expression status) in MISTRG6 mice. This data is now presented in Fig EV3D and E along with the following text:

Results, p.10: The primary lymphoma cells used here express high levels of the IL-6R α -chain and of the signaling chain gp130 and are positive for the phosphorylated form of the signal transducer and activator of transcription STAT3, which mediates signal transduction downstream of the IL-6R heterodimer (Fig 4G and H). The expression of both chains of the IL-6R could further also be detected by qRT-PCR of immunomagnetically enriched CD19⁺ B-cells from this patient's bone marrow aspirate (Fig EV3D and E). To address whether the expression of the IL-6R would generally correlate with the propensity for engraftment of primary human DLBCL cells, we subjected three additional patient samples to immunomagnetic enrichment followed by qRT-PCR for both IL-6R chains, and transplanted all available cells from each donor (in some cases, this number was under 100,000 cells) into MISTRG6 mice. Of the three additional examined primary cell samples, only one tested positive for both IL-6R chains at the transcript level (Fig EV3D and E). None of the three showed evidence of engraftment in the spleen or bone marrow of MISTRG6 mice in the examined time frame of eight weeks (Fig EV3E). These results indicate that human IL-6 may promote the orthotopic engraftment and growth of primary DLBCL cells; expression of a functional heterodimeric IL-6R likely is necessary, but not sufficient for successful engraftment. Once established, primary DLBCL cells can readily be passaged in MISTRG6 mice without losing the characteristics of the human patient material.

Materials and Methods, p.24: For qRT-PCR-based determination of gp130 and IL-6RA expression, bone marrow MNCs containing DLBCL cells were enriched for human B cells using the MagniSort™ Human CD19 Positive Selection Kit (Thermo) and then subjected to RNA extraction and qRT-PCR as described in the supplemental materials and methods.

Referee #3 (Remarks for Author):

In the following manuscript, Hashwah and Bertram et. al. examine the role of IL-6 signaling in DLBCL pathogenesis, evaluate its prognostic power, and propose IL-6 targeting as a potential therapeutic strategy for lymphomagenesis. The authors first assessed the ability of DLBCL cell lines to engraft into genetically humanized animal strain MISTRG, that expresses human cytokines (M-CSF, IL-3, GM-CSF, THPO) and hSIRP α in the Rag $^{-/-}$ IL-2R $\gamma^{-/-}$ background and find that although lymphoma cell lines reconstituted the animals, engraftment efficiency was significantly better if MISTRG animals have been previously reconstituted with hematopoietic stem cells or expressed human IL-6 (MISTRG6). Importantly, the authors demonstrate that MISTRG animals reconstituted with human hematopoietic cells and MISTRG6 hosts support engraftment of primary DLBCL samples, a significant finding as it expands scientific toolkit to study primary lymphoma cells. Next, the authors touched up on the mechanisms of STAT3 activation on DLBCL animals and prognostic value of pSTAT3. Finally, the group assessed therapeutic potential of tocilizumab, a neutralizing antibody targeting IL6R and showed that animals treated with tocilizumab showed a significant reduction in lymphoma burden.

I find the studies using MISTRG6 animals and DLBCL xenografts informative and believe authors' findings are highly beneficial to the field overall. My main concern is apparent lack of IL6R surface expression on majority of examined DLBCL cell lines, suggesting that the effect of IL-6 on lymphoma cells, the authors describe, might not be a direct. My second concern is the authors' claims regarding STAT3 activation in different DLBCL samples. A considerably more in-depth analysis is needed to support conclusions the authors make. Overall, I believe the manuscript will significantly benefit if the authors expand on the findings detailing expansion of primary DLBCL cells in MISTRG6 animals and demonstrate surface levels of IL6R on multiple primary DLBCL samples.

General Points:

1. Fig 4G. Please include analysis of IL6R surface levels on at least three primary DLBCL samples.
2. Please clearly indicate how many different primary xenografts were tested for engraftment in MISTRG6 animals.

These two comments raise an important point. We are now explaining in more detail how the expression of IL-6Ra and gp130 affect engraftment of primary DLBCL cells. Indeed, we examined four different primary samples for their ability to engraft in MISTRG6 mice, and also for their expression of both chains of the IL-6R. This information and data are now included in the text and in Fig EV3 as follows:

Results, p.10: The primary lymphoma cells used here express high levels of the IL-6R a-chain and of the signaling chain gp130 and are positive for the phosphorylated form of the signal transducer and activator of transcription STAT3, which mediates signal transduction downstream of the IL-6R heterodimer (Fig 4G and H). The expression of both chains of the IL-6R could further also be detected by qRT-PCR of immunomagnetically enriched CD19 $^{+}$ B-cells from this patient's bone marrow aspirate (Fig EV3D and E). To address whether the expression of the IL-6R would generally correlate with the propensity for engraftment of primary human DLBCL cells, we subjected three additional patient samples to immunomagnetic enrichment followed by qRT-PCR for both IL-6R chains, and transplanted all available cells from each donor (in some cases, this number was under 10.000 cells) into MISTRG6 mice. Of the three additional examined primary cell samples, only one tested positive for both IL-6R chains at the transcript level (Fig EV3D and E). None of the three showed evidence of engraftment in the spleen or bone marrow of MISTRG6 mice in the examined time frame of six weeks (Fig EV3E). These results indicate that human IL-6 may promote the orthotopic engraftment and growth of primary DLBCL cells; expression of a functional heterodimeric IL-6R likely is necessary, but not sufficient for successful engraftment. Once

established, primary DLBCL cells can readily be passaged in MISTRG6 mice without losing the characteristics of the human patient material.

Materials and Methods, p.24: For qRT-PCR-based determination of gp130 and IL-6RA expression, bone marrow MNCs containing DLBCL cells were enriched for human B cells using the MagniSort™ Human CD19 Positive Selection Kit (Thermo) and then subjected to RNA extraction and qRT-PCR as described in the supplemental materials and methods.

3. Figure 7 E. Please perform statistical analysis.

We have performed the Mantel-Cox log rank test for statistical comparison of the three groups. None of the comparisons gave a p-value under 0.05 after correction for multiple comparisons. This is now more explicitly stated in the text.

Results, p.14: However, the majority of mice harboring *MYC* developed lymphomas in their axillary and inguinal lymph nodes and spleen, which was **somewhat (albeit not significantly)** delayed by the genetic ablation of both copies of the *Il6ra* (Figure 7A). Heterozygous mice with one functional copy of *Il6ra* behaved like wild type mice in this setting (Figure 7A). IL-6ra deficiency in B-cells resulted not only in a delay, but also in a reduction of the tumor burden, which manifested in **significantly** smaller lymph nodes at the study end point (Figure 7B).

4. Please increase the font size in the figures. It is extremely difficult to read the labels.

We increased the font size throughout so that it now conforms with the specifications of the journal.

Specific points:

1. The hIL6R expression profiles of RC-K8 and U-2932 cells appears to be duplicated between Figure 3F and Figure 5A. Please remove the duplication and clearly indicate how many times the staining was done.

The staining was performed at least twice and up to four times depending on the cell line. Different FACS plots are now shown in the two figures, which come from two different biological samples.

Figure legends, p.41: Figure 5. The IL-6 receptor is expressed on a subset of DLBCL. (A-C) The expression of the IL-6Ra (CD126) and signaling chains (gp130) was assessed by flow cytometry (A) and SYBR or Taqman qRT-PCR (B, C) respectively, on a panel of 11 DLBCL cell lines. ABC- and GCB-DLBCL cell lines are color-coded in red and green throughout; RC-K8 are unclassified (grey). **The FACS plots in A are representative of at least two and up to four independent stainings per cell line.**

2. Please add scales bars to figures 4C and make sure that the scale bars (and the numerical designation associated with them) are visible.

This has been taken care of throughout.

2nd Editorial Decision

8 August 2019

Thank you for the submission of your revised manuscript to EMBO Molecular Medicine, and please accept my apologies for the delay in getting back to you, which is due to the fact that referee #1 did not provide his/her report so far. To avoid delaying the process any longer, we made a decision based on the two reports we have.

As you will see, the reviewers are now supportive, and I am pleased to inform you that we will be able to accept your manuscript pending the following final editorial amendments:

***** Reviewer's comments *****

Referee #2 (Remarks for Author):

Interesting study and should be published.

Referee #3 (Comments on Novelty/Model System for Author):

The authors addressed all my comments.

2nd Revision - authors' response

22 August 2019

The authors addressed all editorial issues.

Corresponding Author Name: Anne Müller

Manuscript Number: EMM-2019-10576